# UNDERSTANDING THE EFFECT OF BIAS IN DEEP ANOMALY DETECTION

## ABSTRACT

Anomaly detection presents a unique challenge in machine learning, due to the scarcity of labeled anomaly data. Recent work attempts to mitigate such problems by augmenting training of deep anomaly detection models with additional labeled anomaly samples. However, the labeled data often does not align with the target distribution and introduces harmful bias to the trained model. In this paper, we aim to understand the effect of a biased anomaly set on anomaly detection. We formally state the anomaly detection problem as a supervised learning task, and focus on the anomaly detector's recall at a given false positive rate as the main performance metric. Given two different anomaly score functions, we formally define their difference in performance as the relative *scoring bias* of the anomaly detectors. Along this line, our work provides two key contributions. We establish the first finite sample rates for estimating the relative scoring bias for deep anomaly detection, and empirically validate our theoretical results on both synthetic and real-world datasets. We also provide extensive empirical study on how a biased training anomaly set affects the anomaly score function and therefore the detection performance on different anomaly classes. Our study demonstrates scenarios in which the biased anomaly set can be useful or problematic, and provides a solid benchmark for future research.

## 1 INTRODUCTION

Anomaly detection (Chandola et al., 2009; Pimentel et al., 2014) trains a formal model to identify unexpected or anomalous instances in incoming data, whose behaviors differ from normal instances. It is particularly useful for detecting problematic events such as digital fraud, structural defects, and system malfunctions. Building accurate anomaly detection models is a well-known challenge in machine learning, due to the scarcity of labeled anomaly data. The classical and most common approach is to train anomaly detection models using only normal data[1], i.e., first train a model using a corpus of normal data to capture *normal* behaviors, then configure the model to flag instances with large deviations as anomalies. Researchers have also developed deep learning methods to better capture the complex structure in the data (Ruff et al. (2018); Wang et al. (2019a); Zhou & Paffenroth (2017)). Following the terminology introduced by Chandola et al. (2009), we refer to these models as semi-supervised anomaly detection.

Recently, a new line of anomaly detection models proposes to leverage available labeled anomalies during model training, i.e., train an anomaly detection model using both normal data and additional labeled anomaly samples as they become available (Ruff et al. (2020b); Yamanaka et al. (2019); Ruff et al. (2020a); Hendrycks et al. (2019a)). Existing works show that these new models achieve considerable performance improvements beyond the models trained using only normal data. We hereby refer to these models as deep supervised[2] anomaly detection (Chandola et al., 2009).

When exploring these models, we found that when the labeled anomalies (used to train the model) do not align with the target distribution, they could introduce harmful bias to the trained model. Specifically, when comparing the performance of a supervised anomaly detector to its semi-supervised

---

[1]Existing literature has used different terms to describe this type of models: some using semi-supervised anomaly detection (Chandola et al., 2009) and others using unsupervised anomaly detection (Ruff et al., 2018).

[2]Some works termed these models as semi-supervised anomaly detection (Ruff et al., 2020b; Yamanaka et al., 2019; Ruff et al., 2020a; Hendrycks et al., 2019a) while others termed them as supervised anomaly detection (Chandola et al., 2009).

version, the performance difference varies significantly across test anomaly data, some better and some worse. That is, using labeled anomalies during model training does not always improve model performance; instead, it may introduce large variance (or bias) in anomaly detection outcomes.

In this paper, we aim to understand the effect of a biased training set on deep anomaly detection models. We formally state the anomaly detection problem, focusing on the anomaly detector's recall at a given false positive rate as the main performance metric. We factor the contribution of the labeled anomalies by the detector's anomaly scoring function, and show that different types of labeled anomalies produce different anomaly scoring functions. Next, given any two different anomaly scoring functions, we formally define their difference in performance as the relative *scoring bias* of the anomaly detectors. Our novel notion of scoring bias for anomaly detection aligns with the notion of bias in the classical supervised learning setting, with the key difference being the different performance metric—we target recall at a given false positive rate, the metric used by real-world anomaly detection tasks (Li et al., 2019; Liu et al., 2018).

Along this line, we establish the first finite sample rates for estimating the relative scoring bias for deep anomaly detection. We empirically validate our assumptions and theoretical results on both synthetic and three real-world datasets (Fashion-MNIST, Statlog (Landsat Satellite), and Cellular Spectrum Misuse (Li et al., 2019)).

Furthermore, we provide an empirical study on how a biased training anomaly set affects the anomaly score function and therefore the resulting detection performance. We consider the above three real-world datasets and six deep-learning based anomaly detection models. Our study demonstrates scenarios in which the biased anomaly set can be useful or problematic, and provides a solid benchmark for future research.

In this paper, we introduce a formal analysis on the effect of a biased training set on deep anomaly detection. Our main contributions are the following:

- We discover the issue of large performance variance in deep anomaly detectors, caused by the use of the biased anomaly set as training data.
- We model the effect of biased training as relative scoring bias, and establish the first finite sample rates for estimating the relative scoring bias of the trained models.
- We conduct empirical experiments to verify and characterize the impact of the relative scoring bias on six popular anomaly detection models, and three real-world datasets.

To the best of our knowledge, our work is the first to formally study the effect of a biased anomaly training set on deep anomaly detection. Our results show both significant positive and negative impacts of these biases, and suggest that model trainers must treat anomalies with additional care. We believe this leads to new opportunities for improving deep anomaly detectors and deserves more attention from the research community.

## 2 RELATED WORK

**Anomaly Detection Models.** While the literature on anomaly detection models is extensive, the most relevant to our work are deep learning based models. Following the terminology used by Chandola et al. (2009), we consider two types of models:

- *Semi-supervised anomaly detection* refers to models trained on only normal data, e.g., Ruff et al. (2018); Sakurada & Yairi (2014); Zhou & Paffenroth (2017);
- *Supervised anomaly detection* refers to models trained on normal data and a small set of labeled anomalies, e.g., Pang et al. (2019); Daniel et al. (2019); Yamanaka et al. (2019); Ruff et al. (2020a;b).

One can also categorize models by their architecture: hypersphere (Ruff et al., 2018; 2020a;b) and autoencoder (or reconstruction) based models (Zhou & Paffenroth, 2017; Yamanaka et al., 2019).

Another line of recent work proposes to use synthetic or auxiliary anomalies to train anomaly detection models (Golan & El-Yaniv (2018); Hendrycks et al. (2019c); Lee et al. (2018); Hendrycks et al. (2019b)), "forcing" the model to learn a more compact representation of the normal data. While the existing work has shown empirically that the choice of abnormal data in training can help detect some unseen abnormal distributions, it does not offer any theoretical explanation for the phe-

nomenon, nor does it consider the counter-cases when additional abnormal data in training hurt the detection performance.

**Bias in Anomaly Detection.** To the best of our knowledge, we are the first to identify the presence of bias caused by an additional labeled anomaly set in deep anomaly detection models, especially when there exists a mismatch between the anomalies present in training and those encountered in testing (as shown in Section 5). Existing work has explored the presence of bias in semi-supervised anomaly detection models when there exists defective *normal* data in training, like outliers and simple-to-reconstruct examples (Tong et al., 2019), or examples with background noise (Liu & Ma, 2019). There is also literature on the bias-variance tradeoff for ensembles of semi-supervised anomaly detection models (Aggarwal & Sathe, 2015; Rayana et al., 2016). But little or no work has been done on the bias of anomaly detection in the supervised setting (i.e., models trained on both normal data and some labeled anomalies). Finally, another line of work in transfer learning has identified the value of additional labeled data in training (Kpotufe & Martinet, 2018; Hanneke & Kpotufe, 2019) and the performance bias on target data by transferring knowledge from a less related source (Wang et al., 2019b; Wu et al., 2020). Yet most work only considered the cases of classification models.

**PAC guarantees for Anomaly Detection.** Despite significant progress on developing theoretical guarantees for classification tasks (Valiant (1984); Kearns et al. (1994)), little has been done for anomaly detection tasks. Siddiqui et al. (2016) first establishes a PAC framework for anomaly detection models using the notion of pattern space; however, it is hard to apply such pattern spaces to deep learning models with complex latent spaces. Liu et al. (2018) proposes a model-agnostic approach to provide the PAC guarantee for anomaly detection performance, by analyzing the convergence for the cumulative distribution of anomaly scores. We follow the basic setting from this line of work to address the convergence of the relative scoring bias. In contrast to prior work, our proof relies on a novel adaption of the key theoretical tool from Massart (1990), which allows us to extend our theory to characterize the notion of scoring bias as defined in Section 3.2.

# 3 PROBLEM FORMULATION

We now formally state the anomaly detection problem. Consider a model class $\Theta$ for anomaly detection, and a (labeled) training set $D$ sampled from a mixture distribution $\mathcal{D}$ over the normal and anomalous instances. In the context of anomaly detection, a model $\theta$ maps each input instance $x$ to a continuous output, which corresponds to anomaly score $s_\theta(x)$. The model further uses a threshold $\tau_\theta$ on the score function to produce a binary label for input $x$.

For a given threshold value $\tau_\theta$, we can define the False Positive Rate (FPR) of the model $\theta$ on the input data distribution as $\text{FPR}(s_\theta, \tau_\theta) = \mathbb{P}\left[s_\theta(x) > \tau_\theta \mid y = 0\right]$, and the True Positive Rate (TPR, a.k.a. Recall) as $\text{TPR}(s_\theta, \tau_\theta) = \mathbb{P}\left[s_\theta(x) > \tau_\theta \mid y = 1\right]$. The FPR and TPR are competing objectives—therefore, a key challenge for anomaly detection algorithms is to identify a configuration of the score, threshold pair $(s_\theta, \tau_\theta)$ that strikes a balance between the two performance metrics. W.l.o.g.[3], in this paper we focus on the following scenario, where the objective is to maximize TPR subject to achieving a target FPR. Formally, let $q$ be the target FPR; we define the optimal anomaly detector as[4]

$$(s_\theta^*, \tau_\theta^*) \in \underset{(s_\theta, \tau_\theta):\theta \in \Theta}{\arg\max} \ \text{TPR}(s_\theta, \tau_\theta) \quad \text{s.t. } \text{FPR}(s_\theta, \tau_\theta) \leq q \quad (3.1)$$

## 3.1 A GENERAL ANOMALY DETECTION FRAMEWORK

Note that the performance metric (namely TPR) in Problem 3.1 is statistics that depends on the entire predictive distribution, and can not be easily evaluated on any single data point. Therefore, rather than directly solving Problem 3.1, practical anomaly detection algorithms (such as OCSVM (Schölkopf et al., 1999), Deep SAD (Ruff et al., 2020b), etc) often rely on a two-stage process: (1)

---

[3]Our results can be easily extended to the setting where the goal is to minimize FPR subject to a given TPR.

[4]This formulation aligns with many contemporary works in deep anomaly detection. For example, Li et al. (2019) show that in real-world anomaly detection problems, it is desirable to detect anomalies with a prefixed low false alarm rate; Liu et al. (2018) formulate the anomaly detection in a similar way, where the goal is to minimize FPR for a fixed TPR.

learning the score function $s_\theta$ from training data via a surrogate loss, and (2) given $s_\theta$ from the previous step, computing the threshold function $\tau_\theta$ on the training data. Formally, given a model class $\Theta$, a training set $D$, a loss function $\ell$, and a target FPR $q$, a two-staged anomaly detection algorithm outputs

$$\begin{cases} \hat{s}_\theta \in \arg\min_{s_\theta : \theta \in \Theta} \ell(s_\theta, D) \\ \hat{\tau}_\theta \in \arg\max_{\tau_\theta : \theta \in \Theta} \text{TPR}(\hat{s}_\theta, \tau_\theta) \quad \text{s.t. FPR}(\hat{s}_\theta, \tau_\theta) \le q \end{cases} \tag{3.2}$$

Note that the first part of Equation 3.2 amounts to solving a supervised learning problem. Here, the loss function $\ell$ could be instantiated into latent-space-based losses (e.g., Deep SAD), margin-based losses (e.g., OCSVM), or reconstruction-based losses (e.g., ABC (Yamanaka et al., 2019)); therefore, many contemporary anomaly detection models fall into this framework. To set the threshold $\hat{\tau}_\theta$, we consider using the distribution of the anomaly scores $\hat{s}_\theta(\cdot)$ from a labeled validation set $D^{\text{val}} \sim \mathcal{D}$. Let $D^{\text{val}} := D_0^{\text{val}} \cup D_a^{\text{val}}$ where $D_0^{\text{val}}$ and $D_a^{\text{val}}$ denote the subset of normal data and the subset of abnormal data of $D^{\text{val}}$. Denote the empirical CDFs for anomaly scores assigned to $x$ in $D_0^{\text{val}}$ and $D_a^{\text{val}}$ as $\hat{F}_0$ and $\hat{F}_a$, respectively. Then, given a target FPR value $q$, following a similar argument as Liu et al. (2018), one can compute the threshold as $\hat{\tau}_\theta = \max\{u \in \mathbb{R} : \hat{F}_0(u) \le q\}$. The steps for solving the second part of Equation 3.2 is summarized in Algorithm 1.

---

**Algorithm 1:** Computing the anomaly detection threshold for Problem 3.2

---

**Data:** A validation dataset $D^{\text{val}}$ and a scoring function $s(\cdot)$.
**Result:** A score threshold achieving a target FPR and the corresponding recall on $D^{\text{val}}$.

1  Get anomaly score $s(x)$ for each x in $D^{\text{val}}$.
2  Compute empirical CDF $\hat{F}_0(x)$ and $\hat{F}_a(x)$ for anomaly scores of $x$ in $D_0^{\text{val}}$ and $D_a^{\text{val}}$.
3  Output detection threshold $\hat{\tau} = \max\{u \in \mathbb{R} : \hat{F}_0(u) \le q\}$.
4  Output TPR (recall) on $D_a^{\text{val}}$ as $\hat{r} = 1 - \hat{F}_a(\hat{\tau})$.

---

### 3.2 Scoring Bias

Given a model class $\Theta$ and a training set $D$, we define the *scoring bias* of a detector $(\hat{s}_\theta, \hat{\tau}_\theta)$ to be the difference in TPR between $(\hat{s}_\theta, \hat{\tau}_\theta)$ and $(s_\theta^*, \tau_\theta^*)$:

$$\text{bias}(\hat{s}_\theta, \hat{\tau}_\theta) := \arg\max_{(s_\theta, \tau_\theta) : \theta \in \Theta} \text{TPR}(s_\theta, \tau_\theta) - \text{TPR}(\hat{s}_\theta, \hat{\tau}_\theta) \tag{3.3}$$

We call $(\hat{s}_\theta, \hat{\tau}_\theta)$ a *biased* detector if $\text{bias}(\hat{s}_\theta, \hat{\tau}_\theta) > 0$. In practice, due to biased training distribution, and the fact that the two-stage process in Equation 3.2 is not directly optimizing TPR, the resulting anomaly detectors are often biased by construction. Therefore, one practically significant performance measure is the *relative bias*, defined as the difference in TPR between two anomaly detectors, subject to the constraints in Equation 3.2. It captures the relative strength of two algorithms in detecting anomalies, and therefore is an important indicator for model evaluation and model selection. Formally, given two *arbitrary* anomaly score functions $s, s'$ and the corresponding threshold function $\tau, \tau'$ obtained from Algorithm 1, we define the *relative scoring bias* between $s$ and $s'$ as:

$$\xi(s, s') := \text{bias}(s, \tau) - \text{bias}(s', \tau') = \text{TPR}(s', \tau') - \text{TPR}(s, \tau) \tag{3.4}$$

Note that when $s' = s_\theta^*$, the relative scoring bias (equation 3.4) reduces to the scoring bias (equation 3.3). We further define the *empirical relative scoring bias* between $s$ and $s'$ as

$$\hat{\xi}(s, s') := \widehat{\text{TPR}}(s', \tau') - \widehat{\text{TPR}}(s, \tau) \tag{3.5}$$

where $\widehat{\text{TPR}}(s, \tau) = \frac{1}{n} \sum_{j=1}^n \mathbf{1}_{s(x_j) > \tau ; y_j = 1}$ denotes the TPR (recall) estimated on a finite validation set of size $n$. In the following sections, we will investigate both the theoretical properties and the empirical behavior of the empirical relative scoring bias for contemporary anomaly detectors.

## 4 Finite Sample Analysis for Empirical Relative Scoring Bias

In this section, we show how one can estimate the relative scoring bias (Equation 3.4) given *any* two scoring functions $s, s'$ learned in Section 3.1. As an example, $s$ could be a scoring function induced

by a semi-unsupervised anomaly detector trained on normal data only, and $s'$ could be a scoring function induced by a supervised anomaly detector trained on biased anomaly set. In the following, we provide a finite sample analysis of the convergence rate of the empirical relative scoring bias, and validate our theoretical analysis via a case study.

## 4.1 FINITE SAMPLE GUARANTEE

**Notations.** Concretely, we assume that when determining the threshold in Line 3 of Algorithm 1, both scoring functions $s, s'$ are evaluated on the unbiased (marginal) empirical distribution of the normal data. Furthermore, the empirical TPR in Line 4 are estimated on the unbiased empirical distribution of the abnormal data. Let $\{s_i := s(x_i) \mid x_i, y_i = 0\}_{i=1}^{n_0}$ denote a set of anomaly scores evaluated by $s(\cdot)$ on $n_0$ *i.i.d.* random normal data points. Following the notation in Section 3.1, we use $F_0(t) := \mathbb{P}[s(x) \le t \mid y = 0]$ to denote the CDF of $s(x)$, and use $\hat{F}_0(t) := \frac{1}{n_0} \sum_{i=1}^{n_0} \mathbf{1}_{s_i \le t; y_i = 0}$ to denote the corresponding empirical CDF. For $n_1$ *i.i.d.* samples $\{s_j := s(x_j) \mid x_j, y_j = 1\}_{j=0}^{n_1}$ with CDF $F_a(t) := \mathbb{P}[s(x) \le t \mid y = 1]$, the corresponding emprical CDF is $\hat{F}_a(t) := \frac{1}{n_1} \sum_{j=1}^{n_1} \mathbf{1}_{s_j \le t; y_j = 1}$. Similarly, we denote the CDF and emiprical CDF for $\{s'_i \mid y_i = 0\}_{i=0}^{n_0}$ as $F'_0(t)$ and $\hat{F}'_0(t)$, and for $\{s'_j \mid y_j = 1\}_{j=0}^{n_1}$ as as $F'_a(t)$ and $\hat{F}'_a(t)$, respectively.

**Infinite sample case.** In the limit of infinite data (both normal and abnormal), $\hat{F}_0, \hat{F}_a, \hat{F}'_0, \hat{F}'_a$ will converge to the true CDFs (cf. Skorokhod's representation theorem and Theorem 2A of Parzen (1980)), and hence the empirical relative scoring bias will also converge. The following Proposition establishes a connection between the CDFs and the scoring bias.

**Proposition 1.** *Given two scoring functions $s, s'$ and a target FPR $q$, the relative scoring bias is*
$$\xi(s, s') = F_a(F_0^{-1}(q)) - F'_a(F_0'^{-1}(q)).$$

Here, $F^{-1}(\cdot)$ is the quantile function. The proof of Proposition 1 follows from the fact that for corresponding choice of $\tau, \tau'$ in Algorithm 1, $\text{TPR}(s, \tau) = 1 - F_a(F_0^{-1}(q))$, and $\text{TPR}(s', \tau') = 1 - F'_a(F_0'^{-1}(q))$.

Next, a direct corollary of the above result shows that, for the special cases where both the scores for normal and abnormal data are Gaussian distributed, one can directly compute the relative scoring bias. The proof is listed in Appendix A.

**Corollary 2.** *Let $q$ be a fixed target FPR. Given two scoring functions $s, s'$, assume that $s(x) \mid (y = 0) \sim \mathcal{N}(\mu_0, \sigma_0)$, $s(x) \mid (y = 1) \sim \mathcal{N}(\mu_a, \sigma_a)$, $s'(x) \mid (y = 0) \sim \mathcal{N}(\mu'_0, \sigma'_0)$, $s'(x) \mid (y = 1) \sim \mathcal{N}(\mu'_a, \sigma'_a)$. Then, the relative scoring bias*
$$\xi(s, s') = \Phi\left(\frac{\sigma_0 \Phi^{-1}(q)}{\sigma_a} + \frac{\mu_0 - \mu_a}{\sigma_a}\right) - \Phi\left(\frac{\sigma'_0 \Phi^{-1}(q)}{\sigma'_a} + \frac{\mu'_0 - \mu'_a}{\sigma'_a}\right)$$
*where $\Phi$ denotes the CDF of the standard Gaussian.*

**Finite sample case.** In practical scenarios, when comparing the performance of two scoring functions $s$ and $s'$, we would only have access to finite samples from the validation set, and hence it is crucial to bound the estimation error due to insufficient samples. We now establish a finite sample guarantee for the estimating the relative scoring bias. Our result extends the analysis of Liu et al. (2018), where we follow the convention to assume that the anomaly data amounts to an $\alpha$ fraction of the mixture distribution. The validation set contains a mixture of $n = n_0 + n_1$ i.i.d. samples, with $n_0$ normal samples and $n_1$ abnormal samples where $\frac{n_1}{n} = \alpha$.

The following result shows that under mild assumptions of the continuity of the CDFs and quantile functions $F_a, F'_a, F_0^{-1}, F_0'^{-1}$, the sample complexity for achieving $|\hat{\xi} - \xi| \le \epsilon$:

**Theorem 3.** *Assume that $F_a, F'_a, F_0^{-1}, F_0'^{-1}$ are Lipschitz continuous with Lipschitz constant $\ell_a, \ell'_a, \ell_0^-, \ell_0'^-$, respectively. Let $\alpha$ be the fraction of abnormal data among $n$ i.i.d. samples from the mixture distribution. Then, w.p. at least $1 - \delta$, with*
$$n \ge \frac{8}{\epsilon^2} \cdot \left(\log \frac{2}{1 - \sqrt{1 - \delta}} \cdot \left(\frac{2 - \alpha}{\alpha}\right)^2 + \log \frac{2}{\delta} \cdot \frac{1}{1 - \alpha}\left(\left(\frac{\ell_a}{\ell_0^-}\right)^2 + \left(\frac{\ell'_a}{\ell_0'^-}\right)^2\right)\right)$$
*the empirical relative scoring bias satisfies $|\hat{\xi} - \xi| \le \epsilon$.*

| Type | Semi-supervised (trained on normal data) | Supervised (trained on normal & some abnormal data) |
|---|---|---|
| Hypersphere-based | Deep SVDD | Deep SAD
Hypersphere Classifier (HSC) |
| Reconstruction-based | Autoencoder (AE) | Semi-supervised AE (SAE)
Autoencoding Binary Classifier (ABC) |

Table 1: The anomaly detection models considered in our case study. Deep SAD and HSC are the supervised versions of Deep SVDD (the semi-supervised baseline model); SAE and ABC are the supervised versions of AE (the semi-supervised baseline model).

We defer the proof of Theorem 3 to Appendix B. Similarly with the open category alien detection setting as discussed in Liu et al. (2018), the sample complexity for estimating the relative scoring bias $n$ grows as $\mathcal{O}\left(\frac{1}{\alpha^2\epsilon^2}\log\frac{1}{\delta}\right)$. Note the analysis of our bound involves a novel two-step process which first bounds the estimation of the threshold for the given FPR, and then leverages the Lipschitz continuity condition to derive the final bound.

## 4.2 CASE STUDY

We conduct a case study to validate our main results above, by training anomaly detection models using a synthetic dataset (Liu et al., 2018) and three real-world datasets. We consider six anomaly detection models listed in Table 1, and they lead to similar results. For brevity, we show results when using Deep SVDD (Ruff et al., 2018) as the baseline model (i.e. trained on normal data only) and Deep SAD (Ruff et al., 2020b) as the semi-supervised model trained on normal and some abnormal data. Later in Appendix D, we include results of other model pairs, including Deep SVDD vs. Hypersphere Classifier (HSC) (Ruff et al., 2020a), Autoencoder (AE) vs. Semi-supervised Autoencoder (SAE)[5], and AE vs. ABC (Yamanaka et al., 2019).

**Our synthetic dataset.** Similar to Liu et al. (2018), we generate our synthetic dataset by sampling data from a mixture data distribution $S$, w.p. $1 - \alpha$ generating the normal data distribution $S_0$ and w.p. $\alpha$ generating the abnormal data distribution $S_a$. Data points in $S_0$ are sampled randomly from a 9-dimensional Gaussian distribution, where each dimension is independently distributed as $\mathcal{N}(0, 1)$. Data points in $S_a$ are sampled from another 9-dimensional distribution, which w.p. 0.4 have 3 dimensions (uniformly chosen at random) distributed as $\mathcal{N}(1.6, 0.8)$, w.p. 0.6 have 4 dimensions (uniformly chosen at random) distributed as $\mathcal{N}(1.6, 0.8)$, and have the remaining dimensions distributed as $\mathcal{N}(0, 1)$. This ensures meaningful feature relevance, point difficulty and variation for the abnormal data distribution as discussed in Emmott et al. (2015).

We obtain two score functions $s$ and $s'$ by training Deep SVDD and Deep SAD respectively on samples from the synthetic dataset (10K data from $S_0$, 1K data from $S_a$). We configure the training, validation and test set so there is no data overlap in them. Thus the training procedure will not affect the sample complexity for estimating the relative scoring bias. To set the anomaly threshold, we fix the target FPR to be 0.05, and vary the number of normal data in the validation set $n$ from {100, 1K, 10K}. We then test the score function and threshold on a fixed test dataset with a large number (20K) of normal data and $\alpha \times$ 20K of abnormal data. We vary $\alpha$ from {0.01, 0.05, 0.1, 0.2}.

**Real-world datasets.** We consider three real-world datasets targeting disjoint subjects: Fashion-MNIST (Xiao et al., 2017) is a collection of images of fashion objects, where we choose some objects as normal and the rest as abnormal; StatLog (Srinivasan, 1993) is a collection of satellite images on various soil types; and Cellular Spectrum Misuse (Li et al., 2019) is a real-world anomaly dataset on cellular spectrum usage, including normal usage and those under four types of attacks. Detailed descriptions of these datasets and training configurations are listed in Appendix C. Like the above, we obtain $s$, $s'$, and anomaly threshold (at a target FPR of 0.05) from these datasets, and test the score function and threshold on their corresponding test datasets and different $\alpha$ values.

**Distribution of anomaly scores.** We first study the distribution of anomaly scores. Figure 1 is a sample plot of score distributions on the test set with $\alpha = 0.1$. We plot the scores for normal and abnormal test data separately, for both scoring functions (derived from Deep SVDD and Deep SAD models respectively). We make two key observations. First, all the distribution curves follow a rough bell shape. Second and more importantly, while the abnormal score distribution closely mimics the normal score distribution under the unsupervised model, it deviates largely from the normal

---

[5]We design SAE by forcing the reconstruction errors to be maximized for additional labeled anomalies encountered in training the autoencoder.

score distribution after semi-supervised learning (i.e., similar mean but much higher variance). This confirms that semi-supervised learning does introduce additional bias in anomaly scores.

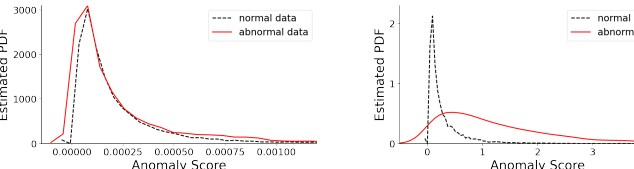

Figure 1: Anomaly score distributions for Deep SVDD (left) and Deep SAD (right) trained and tested using the synthetic dataset, estimated by Kernel Density Estimation.

We also examine the anomaly score distributions for models trained on real-world anomaly detection data sets, including Fashion-MNIST and Cellular Spectrum Misuse. While the score distributions are less close to Gaussian, we do observe the same trend where normal and abnormal score distributions become significantly different after applying semi-supervised learning. The results are shown in Figure 7 and 8 in Appendix D.

**Convergence of relative scoring bias ($\hat{\xi}$) and FPR.** Next we examine the convergence of the empirical FPR obtained from $s'$ (semi-supervised model) and the empirical relative scoring bias $\hat{\xi}$ (computed as the difference of the empirical TPR according to Equation 3.5) obtained from $s$ (semi-supervised model with normal only) and $s'$ (supervised model with biased anomaly). Here we present the convergence results in Figure 2 for the synthetic dataset in terms of the quantile distribution of $\hat{\xi}$ between Deep SVDD (semi-supervised) and Deep SAD (supervised) and the quantile distribution of Deep SAD's FPR. Results for other models and three real-world datasets are in Appendix D, and show consistent trends.

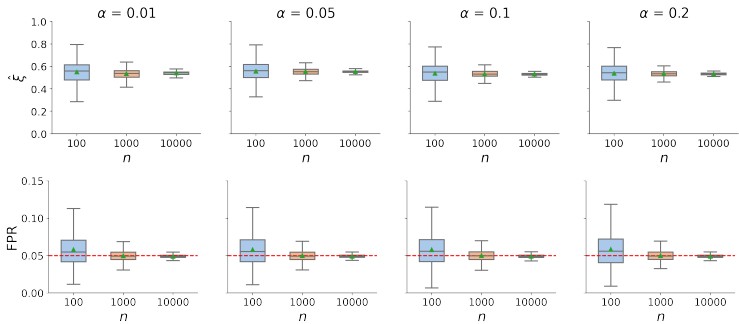

Figure 2: Models (Deep SVDD v.s. Deep SAD) trained on the synthetic dataset: the quantile distribution of relative scoring bias $\hat{\xi}$ (top 4 figures) and FPR (bottom 4 figures), computed on the test set over 1500 runs. $n$ =100, 1000 or 10000; $\alpha$ =0.01, 0.05, 0.1, 0.2. The triangle in each boxplot is the mean. For FPR, the red dotted line marks the target FPR of 0.05.

Similar to our theoretical results, we observe a consistent trend of convergence in FPR and $\hat{\xi}$ as the sample complexity goes up. More specifically, for FPR, as $n$ goes up, it converges to the prefixed value of $0.05$; for $\hat{\xi}$, it also converges to a certain level.

We also examine the rate of convergence w.r.t to $n$. Section 4.1 shows that $n$ required for estimating $\hat{\xi}$ grows in the same order as $\frac{1}{\alpha^2\epsilon^2} \log \frac{1}{\delta}$. That is, the estimation error $\epsilon$ decreases at the rate of $\frac{1}{\sqrt{n}}$; furthermore, as $\alpha$ increases, $n$ required for estimating $\hat{\xi}$ decreases. This can be seen from Figure 2 (top figure) where at $n = 10000$, the variation of $\hat{\xi}$ at $\alpha = 0.2$ is 50% less than that at $\alpha = 0.01$.

## 5 IMPACT OF SCORING BIAS ON ANOMALY DETECTION PERFORMANCE

We perform empirical experiments to study the end-to-end impact of relative scoring bias on deep anomaly detection models. Our goal is to understand the type and severity of performance variations introduced by different anomaly training sets.

**Experiment setup.**    We consider six deep anomaly detection models previously listed in Table 1, and three real-world datasets: Fashion-MNIST, Statlog (Landsat Satellite) and Cellular Spectrum Misuse. For each dataset, we build normal data by choosing a single class (e.g., `top` in Fashion-MNIST, `normal` in Cellular Spectrum Misuse), and treat the other classes as the abnormal classes. Note that Cellular Spectrum Misuse is a real-world anomaly dataset where the abnormal classes are attacks against today's cellular networks (Li et al., 2019).

From those abnormal classes, we pick a single class as the abnormal training data, and the rest as the abnormal test data on which we test separately. Given the data, we train $\theta_0 := (s_{\theta_0}, \tau_{\theta_0})$, a semi-supervised anomaly detector using normal training data, and $\theta_s := (s_{\theta_s}, \tau_{\theta_s})$ a supervised anomaly detector using both normal and abnormal training data (with a 10:1 normal vs. anomaly ratio). We follow the original paper of each model to implement the model and its training. For each trained model, we configure the anomaly score threshold to reach a target false positive rate (FPR) of 0.05. We then test these trained models against various abnormal test classes, and record the recall (TPR) value for each abnormal test class. We repeat the above by selecting different abnormal training data. Detailed descriptions of these datasets and training configurations are listed in Appendix C.

We evaluate the potential bias introduced by different abnormal training data by comparing the model recall (TPR) value of both $\theta_0$ and $\theta_s$ against different abnormal test data. We define the bias to be upward (↑) if $\text{TPR}(\theta_s) > \text{TPR}(\theta_0)$, and downward (↓) if $\text{TPR}(\theta_s) < \text{TPR}(\theta_0)$.

We group our experiments into three scenarios: (1) when abnormal training data is visually similar to normal training data; (2) when abnormal training data is visually dissimilar to normal training data; and (3) when abnormal training data is a weighted combination of (1) and (2). Here we compute visual similarity as the $L^2$ distance. The similarity results are listed in Appendix E.

We observe similar trends across all three datasets and all six anomaly detection models. For brevity, we summarize our observations below, and further illustrate them using examples that consider two models (Deep SVDD as $\theta_0$ and Deep SAD as $\theta_s$), and two datasets (Fashion-MNIST, Cellular Spectrum Misuse). We list full results (mean/std) on all the models and datasets in Appendix E.

**Scenario 1: Abnormal training data visually similar to normal training data.**    In this scenario, the use of abnormal data in model training (or supervised model) does improve detection of abnormal data in the training class, but also creates considerable performance changes, both upward and downward, for other classes of abnormal test data. The direction of change depends heavily on the similarity of the abnormal test data to the training abnormal data. The model performance on test data similar to the training abnormal data moves *upward* significantly while that on test data dissimilar to the training abnormal moves *downward* significantly.

We illustrate this observation using examples of Fashion-MNIST and Cellular Spectrum Misuse. For Fashion-MNIST, the normal and abnormal training classes are `top` and `shirt`, respectively, which are similar to each other. Figure 3(a) plots the recalls of model $\theta_0$ and $\theta_s$ for all abnormal test classes, arranged by a decreasing similarity to the training abnormal class (`shirt`). We see that $\text{TPR}(\theta_s)$ on classes similar to `shirt` (including itself) is significantly higher than $\text{TPR}(\theta_0)$ (e.g. increased from <0.2 to 0.9 for `pullover`). But for classes dissimilar from `shirt`, $\text{TPR}(\theta_s)$ is either similar or significantly lower (e.g., reduced from 0.9 to 0.4 for `boot`). For Cellular Spectrum Misuse, the normal and abnormal training classes are `normal` and `NB-10ms`, respectively. The effect of training bias is highly visible in Figure 3(c), where $\text{TPR}(\theta_s)$ on `NB-10ms` and `NB-5ms` rises from almost zero to >93% while $\text{TPR}(\theta_s)$ on `WB-nlos` and `WB-los` drops by 50% or more.

**Scenario 2: Abnormal training data visually dissimilar to normal training data.**    Like Scenario 1, the use of abnormal training does improve the detection of abnormal data belonging to the training class and those similar to the training class. But different from Scenario 1, we observe very little downward changes at abnormal classes dissimilar to the training abnormal.

This is illustrated using another Fashion-MNIST example in Figure 3(b). While the normal training class is still `top`, we use a new abnormal training class of `sneaker` that is quite dissimilar from `top`. We see that $\text{TPR}(\theta_s)$ on `sneaker`, `sandal`, `boot` and `bag` are largely elevated to 0.8 and higher, while $\text{TPR}(\theta_s)$ on other classes are relatively stable (except for `trouser` which more than doubles). Finally, the same applies to another example of Cellular Spectrum Misuse in Figure 3(d) where the abnormal training class is `WB-los`, which is quite different from the normal data. In this case, we observe little change to the model recall.

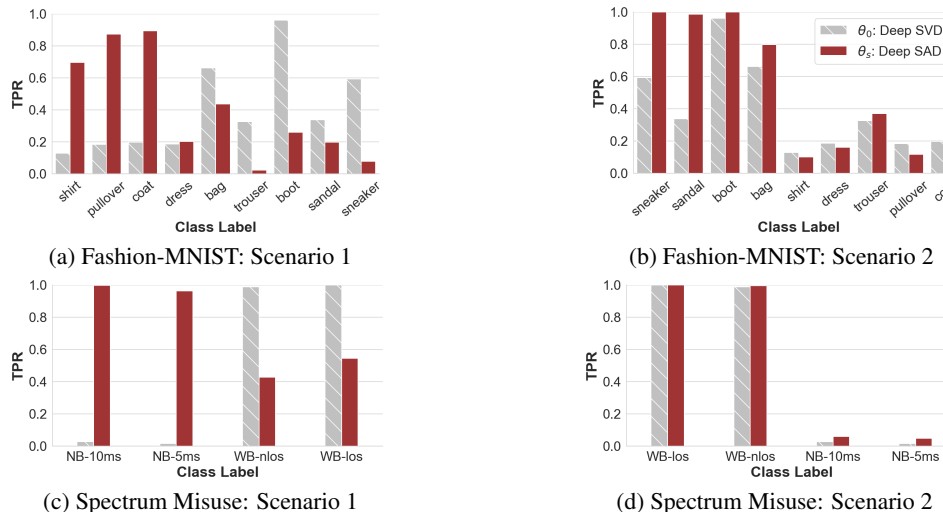

Figure 3: Model TPR under Scenario 1 and 2, trained on two real-world datasets: Fashion-MNIST and Cellular Spectrum Misuse. In each figure, we compare the performance of $\theta_0$ = Deep SVDD (semi-supervised) and $\theta_s$ = Deep SAD (supervised) when tested on abnormal data. We arrange abnormal test data (by their class label) in decreasing similarity with training abnormal data. The leftmost entry in each figure is the class used for abnormal training. For Fashion-MNIST, the normal data is `top`; for Cellular Spectrum Misuse, the normal data is `normal`.

**Scenario 3: Mixed abnormal training data.** We run three configurations of group training on Fashion-MNIST (normal: `top`; abnormal: `shirt` & `sneaker`) by varying the weights of the two abnormal classes in training (0.5/0.5, 0.9/0.1, 0.1/0.9). The detailed results for each weight configuration are listed in Appendix E. Overall, the use of group training does improve the model performance. However, under all three weight configurations, we observe a consistent pattern of downward bias for an abnormal test class (`trouser`) and upward bias for most other abnormal classes. Note that `trouser` is relatively more dissimilar to both training abnormal classes.

**Summary of observations.** Our empirical study shows that a biased (anomaly) training set can introduce significant impact on deep anomaly detection, especially on *whether the use of labeled anomalies in training would help detect unseen anomalies*. When the labeled anomalies are similar to the normal instances, the trained model will likely face large performance degradation on unseen anomalies *different* from the labeled anomalies, but improvement on those *similar* to the labeled anomalies. Yet when the labeled anomalies are dissimilar to the normal instances, the supervised model is more useful than its semi-supervised version. Such difference in model behavior is likely because different types of abnormal training data affect the training distribution (thus the scoring function) differently. In particular, when the labeled anomalies are similar to the normal data, they lead to large changes to the scoring function and affect the detection of unseen anomalies "unevenly". Overall, our results suggest that model trainers must treat labeled anomalies with care.

## 6 CONCLUSIONS AND FUTURE WORK

To the best of our knowledge, our work provides the first formal analysis on how a biased anomaly training set affects deep anomaly detection. We define and formulate its impact on anomaly detector's recall (or TPR) as the relative *scoring bias* of the detector when comparing to its semi-supervised baseline trained on only normal data. We then establish the first finite sample rates for estimating the relative scoring bias for supervised anomaly detection, and empirically validate our theoretical results on both synthetic and real-world datasets. We also empirically study how such relative scoring bias translates into variance in detector performance against different unseen anomalies, and demonstrate scenarios in which the biased anomaly set can be useful or harmful. Our work exposes a new challenge in training deep anomaly detection models, especially when labeled abnormal data becomes available. An open question is how to construct an unbiased anomaly detector, even when having access to the true anomaly distribution. As future work, we plan to develop new training procedures that can leverage labeled anomalies to exploit upward scoring bias while avoiding downward scoring bias.

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

## A PROOF OF COROLLARY 2

*Proof of Corollary 2.* Assuming the score functions are Gaussian distributed, we can denoted $F_0(s)$ as $\Phi(\frac{s-\mu_0}{\sigma_0})$, $\tilde{F}_0(s)$ as $\Phi(\frac{s-\tilde{\mu}_0}{\tilde{\sigma}_0})$, $F_a(s)$ as $\Phi(\frac{s-\mu_a}{\sigma_a})$, and $\tilde{F}_a(s)$ as $\Phi(\frac{s-\tilde{\mu}_a}{\tilde{\sigma}_a})$.

Therefore, we have $\Delta_0 = |(\sigma_0\Phi^{-1}(q) + \mu_0) - (\tilde{\sigma}_0\Phi^{-1}(q) + \tilde{\mu}_0)|$.

Thus,

$$
\begin{aligned}
\xi_N &:= \tilde{r} - r \\
&= F_a(F_0^{-1}(q)) - \tilde{F}_a(F_0^{\tilde{-}1}(q)) \\
&= \Phi(\frac{\sigma_0\Phi^{-1}(q)}{\sigma_a} + \frac{\mu_0 - \mu_a}{\sigma_a}) - \Phi(\frac{\tilde{\sigma}_0\Phi^{-1}(q)}{\tilde{\sigma}_a} + \frac{\tilde{\mu}_0 - \tilde{\mu}_a}{\tilde{\sigma}_a})
\end{aligned}
$$

$\square$

## B PROOF OF THEOREM 3

*Proof of Theorem 3.* Our proof builds upon and extends the analysis framework of Liu et al. (2018), which relies on one key result from Massart (1990),

$$
\mathbb{P}\left[\sqrt{n}\sup_x |\hat{F}(x) - F(x)| > \lambda\right] \leq 2\exp(-2\lambda^2). \tag{B.1}
$$

Here, $\hat{F}(x)$ is the empirical CDF calculated from $n$ samples. Given a fixed threshold function, Liu et al. (2018) showed that it required

$$
n > \frac{1}{2\epsilon_1^2} \log \frac{2}{1 - \sqrt{1-\delta}} \cdot \left(\frac{2-\alpha}{\alpha}\right)^2 \tag{B.2}
$$

examples, in order to guarantee $|\hat{F}_a(x) - F_a(x)| \leq \epsilon_1$ with probability at least $1 - \delta$ (recall that $\alpha$ denotes the fraction of abnormal data among the $n$ samples).

Here we note that our proof relies on a novel adaption of Equation B.1, which allows us to extend our analysis to the convergence of quantile functions.

To achieve this goal, we further assume the Lipschitz continuity for the CDFs/quantile functions:

$$
|F_a(x) - F_a(x')| \leq \ell_a|x - x'| \tag{B.3}
$$
$$
|F_a'(x) - F_a'(x')| \leq \ell_a'|x - x'| \tag{B.4}
$$
$$
|F_0^-(x) - F_0^-(x')| \leq \ell_0^-|x - x'| \tag{B.5}
$$
$$
|F_0'^-(x) - F_0'^-(x')| \leq \ell_0'^-|x - x'| \tag{B.6}
$$

Combining the above inequalities equation B.5 with equation B.1, we obtain

$$
\mathbb{P}\left[\sup_q \left|\hat{F}_0^{-1}(q) - F_0^{-1}(q)\right| \geq \frac{\lambda}{\sqrt{n_0}}\right] \leq \mathbb{P}\left[\sup_q \left|F_0\left(\hat{F}_0^{-1}(q)\right) - F_0\left(F_0^{-1}(q)\right)\right| \geq \frac{\lambda}{\sqrt{n_0}\ell_0^-}\right] \tag{B.7}
$$

Let $q = \hat{F}_0(x)$, then equation B.7 becomes

$$
\begin{aligned}
&\mathbb{P}\left[\left|F_0\left(\hat{F}_0^{-1}(\hat{F}_0(x))\right) - F_0\left(F_0^{-1}(\hat{F}_0(x))\right)\right| \geq \frac{\lambda}{\sqrt{n_0}\ell_0^-}\right] \\
=&\mathbb{P}\left[\left|F_0(x) - \hat{F}_0(x))\right| \geq \frac{\lambda}{\sqrt{n_0}\ell_0^-}\right] \\
\leq& 2e^{-\frac{2}{n_0}\left(\frac{\lambda}{\ell_0^-}\right)^2}
\end{aligned} \tag{B.8}
$$

Therefore, in order for $\mathbb{P}\left[\sup_q \left|\hat{F}_0^{-1}(q) - F_0^{-1}(q)\right| \geq \epsilon_2\right] \leq \delta$ to hold, it suffices to set

$$n_0 = n(1 - \alpha) > \frac{1}{2}\frac{1}{\epsilon_2^2}\frac{1}{\ell_0^{-2}} \log \frac{2}{\delta} \tag{B.9}$$

Furthermore, combining equation B.1, equation B.2, and equation B.3, we get

$$\hat{F}_a(\tau_n) \leq F_a(\tau) + (\tau - \tau_n)\ell_a + \epsilon_1 \tag{B.10}$$

Subtitute $\tau_n = \hat{F}_0^{-1}(q)$, and $\tau = F_0^{-1}(q)$ in the above inequality, and set $\epsilon_1 = \frac{\epsilon}{4}, \epsilon_2 = \frac{\epsilon}{4\ell_a}$, we get

$$\left|\hat{F}_a\left(\hat{F}_0^{-1}(q)\right) - F_a\left(F_0^{-1}(q)\right)\right| \leq \epsilon/2. \tag{B.11}$$

Similary, we repeat the same procedure for $s'$, and can get

$$\left|\hat{F}'_a\left(\hat{F}_0^{'-1}(q)\right) - F'_a\left(F_0^{'-1}(q)\right)\right| \leq \epsilon/2. \tag{B.12}$$

with probability at least $1 - \delta$.

Therefore, with $n \geq \frac{8}{\epsilon^2} \cdot \left(\log\frac{2}{1-\sqrt{1-\delta}} \cdot \left(\frac{2-\alpha}{\alpha}\right)^2 + \log\frac{2}{\delta} \cdot \frac{1}{1-\alpha}\left(\left(\frac{\ell_a}{\ell_0^-}\right)^2 + \left(\frac{\ell'_a}{\ell_0^{'-}}\right)^2\right)\right)$ examples we can get $|\hat{\xi} - \xi| \leq \epsilon$ with probability at least $1 - \delta$.

$\square$

## C  THREE REAL-WORLD DATASETS AND TRAINING CONFIGURATIONS

**Fashion-MNIST.**  This dataset (Xiao et al. (2017)) is a collection of 70K grayscale images on fashion objects (a training set of 60K examples and a test set of 10K examples), evenly divided into 10 classes (7000 images per class). Each image is of 28 pixels in height and 28 pixels in width, and each pixel-value is an integer between 0 and 255. The 10 classes are denoted as `top`, `trouser`, `pullover`, `dress`, `coat`, `sandal`, `shirt`, `sneaker`, `bag`, `boot`.

To train the anomaly detection models, we pick one class as the normal training class, another class as the abnormal training class, and the rest as the abnormal testing class. We use the full training set of the normal class (6K), and a random 10% of the training set of the abnormal training class (600) to train the deep anomaly detection models. We use the test data of the normal class (1K) to configure the anomaly scoring thresholds to meet a 5% false positive rate (FPR). We then test the models on the full data of each abnormal testing class, as well as on the untrained fraction of the abnormal training class.

**StatLog (Landsat Satellite).**  This dataset (Srinivasan (1993)) is a collection of 6,435 NASA satellite images, each of $82 \times 100$ pixels, valued between 0 and 255. The six labeled classes are denoted as `red soil`, `cotton crop`, `grey soil`, `damp grey soil`, `soil with vegetation stubble`, and `very damp grey soil`. Unless specified otherwise, we follow the same procedure to train the models. The normal training data includes 80% data of the designated class, and the abnormal training data is 10% of the normal training data in size. Due to the limited amount of the data, we use the full data of the normal data to configure the anomaly scoring thresholds to meet a 5% FPR. We then test the models on the full data of each abnormal testing class, as well as the untrained fraction of the abnormal training class.

**Cellular Spectrum Misuse.**  This real-world anomaly dataset measures cellular spectrum usage under both normal scenarios and in the presence of misuses (or attacks) (Li et al., 2019). We obtained the dataset from the authors of Li et al. (2019). The dataset includes a large set (100K instances) of real cellular spectrum measurements in the form of *spectrogram* (or time-frequency pattern of the received signal). Each spectrogram (instance) is a $125 \times 128$ matrix, representing the signal measured over 125 time steps and 128 frequency subcarriers. The dataset includes five classes: `normal` (normal usage in the absence of misuse) and four misuse classes: `WB-los` (wideband attack w/o blockage), `WB-nlos` (wideband attack w/ blockage), `NB-10ms` (narrowband attack) and `NB-5ms` (narrowband attack with a different signal). The sample size is 60K for `normal` and 10K for each abnormal class. To train the models, we randomly sample 20K instances from `normal`, 2K instances from one abnormal class, and configure the anomaly score thresholds to meet a 5% FPR.

# D ADDITIONAL EXPERIMENT RESULTS OF SECTION 4.2

**Anomaly score distributions for models trained on the synthetic dataset.** Figure 4–6 plot the anomaly score distributions for semi-supervised (left figure) and supervised (right figure) models, trained on the synthetic dataset ($\alpha = 0.1$, $n = 10000$), estimated by Kernel Density Estimation.

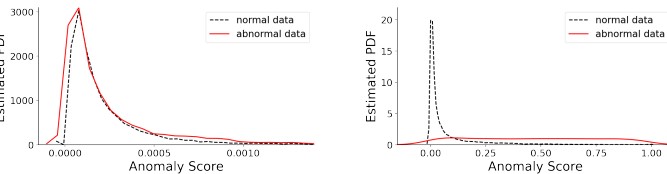

Figure 4: Anomaly score distributions for Deep SVDD (left) and HSC (right), trained on the synthetic dataset.

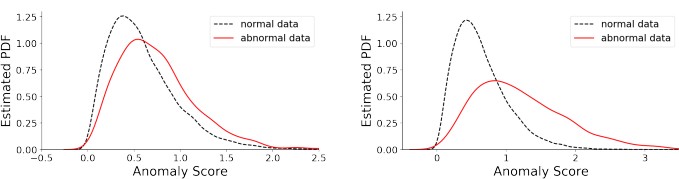

Figure 5: Anomaly score distributions for AE (left) and SAE (right), trained on the synthetic dataset.

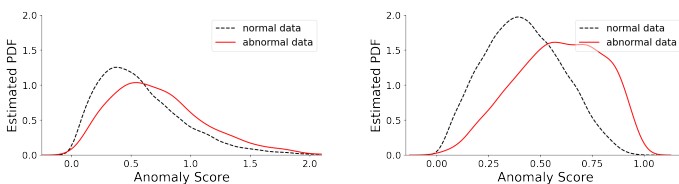

Figure 6: Anomaly score distributions for AE (left) and ABC (right),trained on the synthetic dataset.

**Anomaly score distributions for models trained on real-world datasets.** Figure 7 and 8 plot the anomaly score distributions for semi-supervised and supervised models, when trained on Fashion-MNIST and Cellular Spectrum Misuse, respectively.

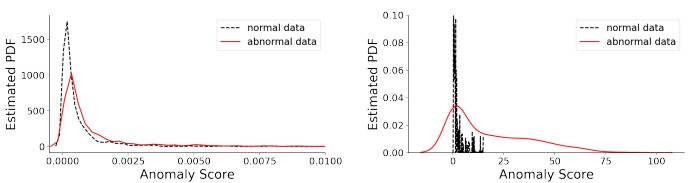

Figure 7: Anomaly score distributions for Deep SVDD (left) and Deep SAD (right), trained on Fashion-MNIST (using `top` as the normal class and `shirt` as the abnormal class).

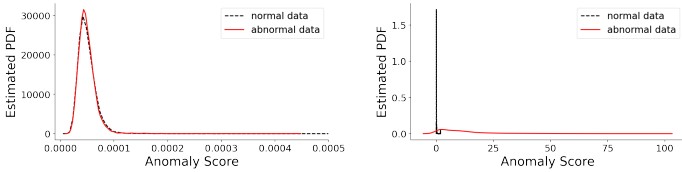

Figure 8: Anomaly score distributions for Deep SVDD (left) and Deep SAD (right), trained on Cellular Spectrum Misuse (using `normal` as the normal class and `NB-10ms` as the abnormal class).

**Convergence of relative scoring bias $\hat{\xi}$ and FRP on the synthetic dataset.** We plot in Figure 9–11 the additional results on (Deep SVDD vs. HSC), (AE vs. SAE), and (AE vs. ABC). Experiment settings are described in Section 4.2. Overall, they show a consistent trend on convergence.

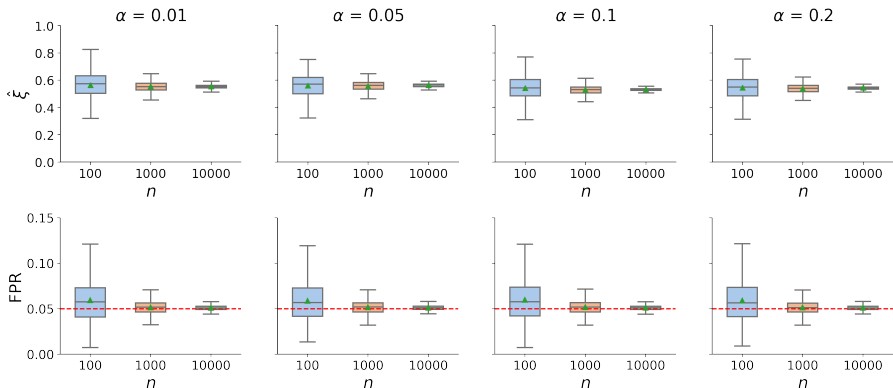

Figure 9: The quantile distribution on the synthetic dataset of (top) relative scoring bias $\hat{\xi}$ and (bottom) FPR, computed on the test set over 1500 runs, for Deep SVDD and HSC.

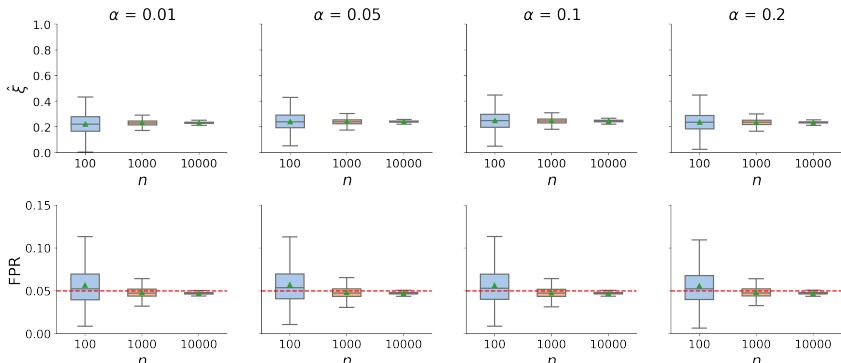

Figure 10: The quantile distribution of (top) relative scoring bias $\hat{\xi}$ and (bottom) FPR, computed on the test set over 1500 runs, for AE and SAE trained on the synthetic dataset.

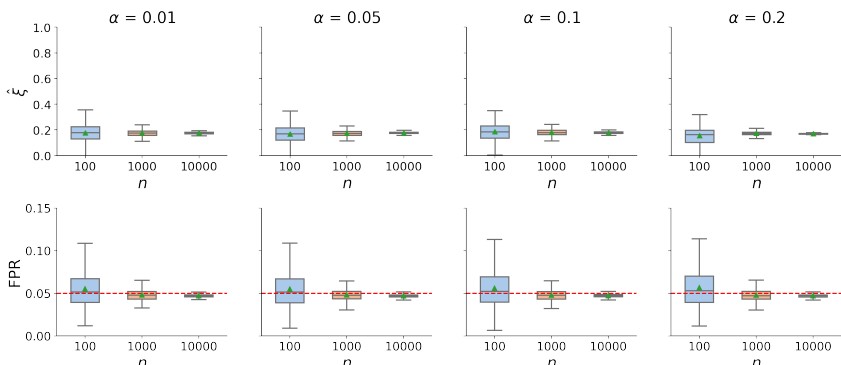

Figure 11: The quantile distribution of (top) relative scoring bias $\hat{\xi}$ and (bottom) FPR, computed on the test set over 1500 runs, for AE and ABC trained on the synthetic dataset.

**Convergence of relative scoring bias $\hat{\xi}$ and FRP on Cellular Spectrum Misuse.** We plot in Figure 12–15 the additional results of (Deep SVDD vs. Deep SAD), (Deep SVDD vs. HSC), (AE vs. SAE), and (AE vs. ABC), when trained on the Cellular Spectrum Misuse dataset. Here we set the normal class as `normal` and the abnormal class as `NB-10ms`, and configure the sample size for the training set as 16K and for the test set as 6K, and vary the sample size for the validation set $n$ from 100, 200, 500, 1K, 2K, 5K. Overall, the plots show a consistent trend on convergence.

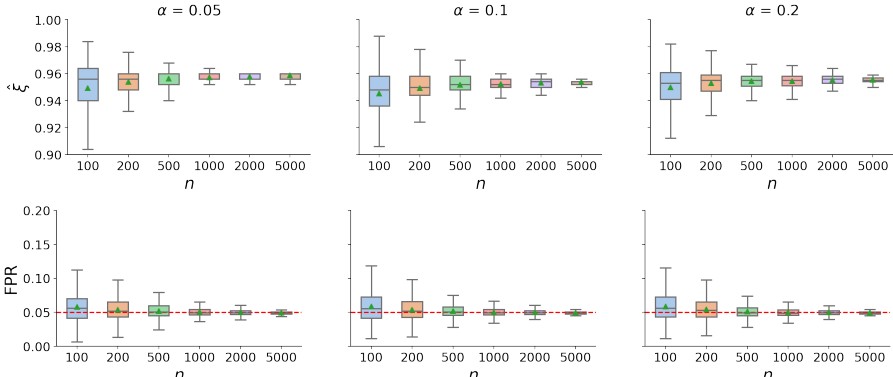

Figure 12: The quantile distribution of relative scoring bias $\hat{\xi}$ (top) and FPR (bottom), computed on the test set over 1000 runs, for Deep SVDD and Deep SAD trained on Cellular Spectrum Misuse.

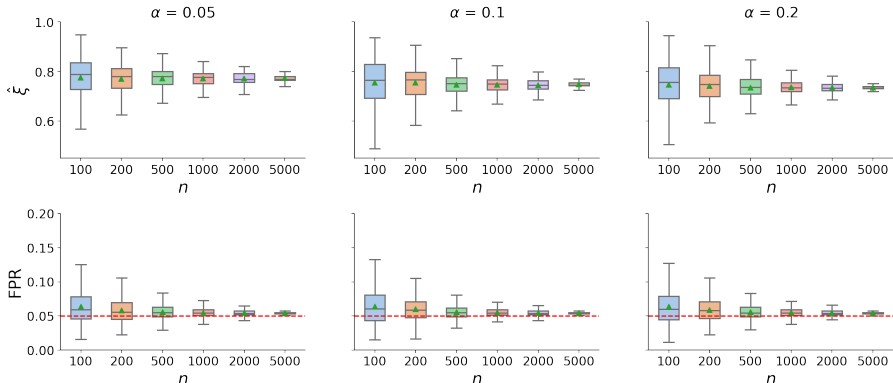

Figure 13: The quantile distribution of (top) relative scoring bias $\hat{\xi}$ and (bottom) FPR, computed on the test set over 1000 runs, for Deep SVDD and HSC trained on Cellular Spectrum Misuse.

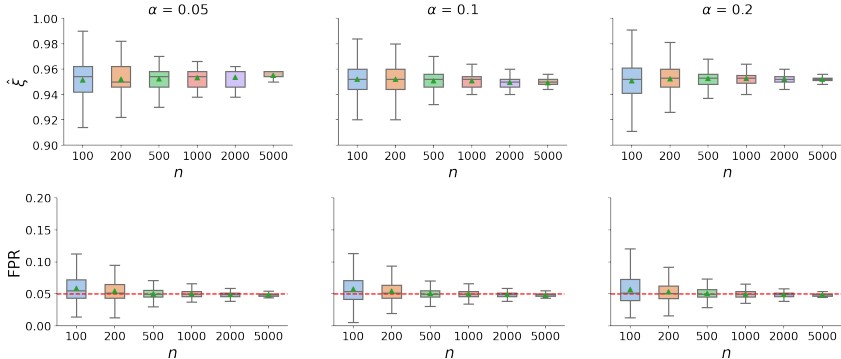

Figure 14: The quantile distribution of (top) relative scoring bias $\hat{\xi}$ and (bottom) FPR, computed on the test set over 1000 runs, for AE and SAE trained on Cellular Spectrum Misuse.

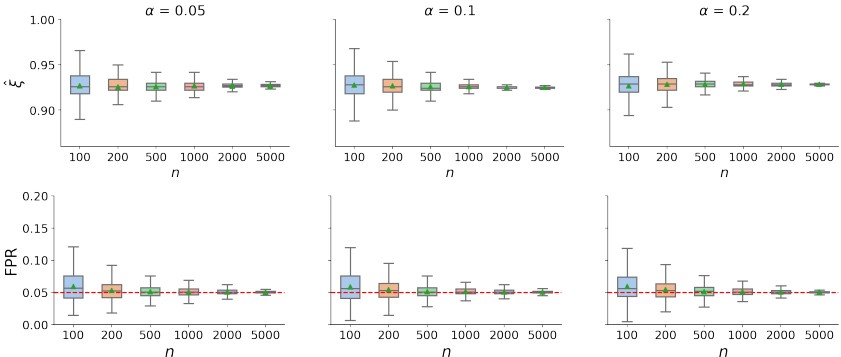

Figure 15: The quantile distribution of (top) relative scoring bias $\hat{\xi}$ and (bottom) FPR, computed on the test set over 1000 runs, for AE and ABC trained on Cellular Spectrum Misuse.

**Convergence of relative scoring bias $\hat{\xi}$ and FRP on FashionMNIST.** Figure 16 plots the convergence of $\hat{\xi}$ and FRP on Deep SVDD vs. Deep SAD models trained on FashionMNIST. The results for other model combinations are consistent and thus omitted. Here we set the normal class as top and the abnormal class as shirt, and configure the sample size for the training set as 3K and for the test set as 2K, and vary the sample size for the validation set $n$ from 100, 200, 500, 1K. Overall, the plots show a consistent trend on convergence.

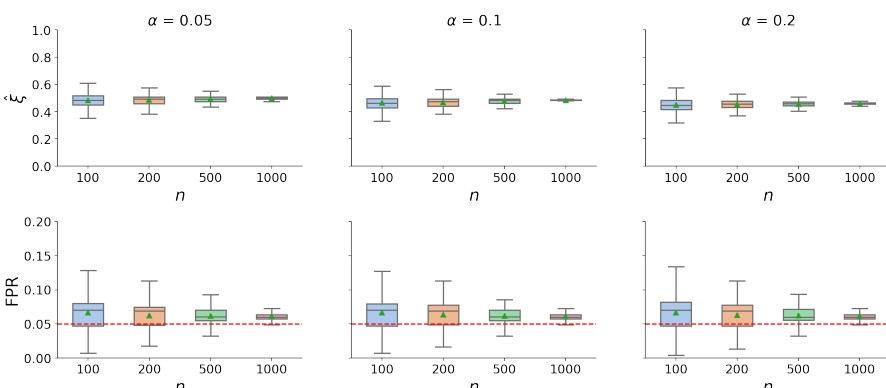

Figure 16: The quantile distribution of (top) relative scoring bias $\hat{\xi}$ and (bottom) FPR, computed on the test set over 100 runs, for Deep SVDD and SAD trained on Fashion-MNIST.

**Convergence of relative scoring bias $\hat{\xi}$ and FRP on StatLog.** Figure 17 plots $\hat{\xi}$ and FRP of Deep SVDD vs. Deep SAD trained on the StatLog dataset. The results for other model combinations are consistent and thus omitted. To maintain a reasonable sample size, we set the normal class to be a combination of `grey soil`, `damp grey soil` and `very damp grey soil`, and the abnormal class to be a combination of `red soil` and `cotton crop`. The sample size for the training is 1.2K and the test set is 1K. We vary the sample size for the validation set $n$ from 100, 200, 500, 1K. Overall, the plots show a consistent trend on convergence.

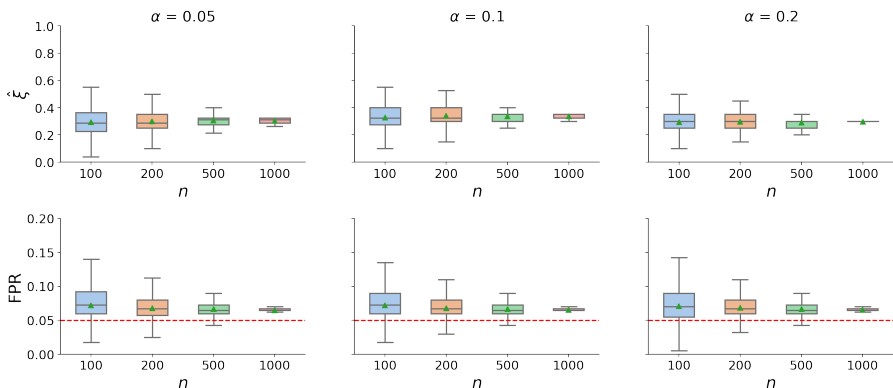

Figure 17: The quantile distribution of (top) relative scoring bias $\hat{\xi}$ and (bottom) FPR, computed on the test set over 100 runs, for Deep SVDD and SAD trained on StatLog.

# E    ADDITIONAL RESULTS OF SECTION 5

**Scenario 1.**    Here the training normal set is visually similar to the training abnormal set. We include the detailed recall result (mean/std over 100 runs) of all six models, and three real-life datasets in Table 2 – 4. Across the six models, the two semi-supervised models (trained on normal data) are Deep SVDD and AE; and the rest are supervised models trained on both normal data and the specified abnormal training data.

In each table, we report the model recall on all abnormal test classes. These abnormal test classes are sorted by decreasing similarity to the abnormal training class (measured by $L^2$, small value = visually similar). Also, ↑ indicates that the supervised model has a higher recall than the semi-supervised model; ↓ indicates the other direction. Overall we observe both upward and downward bias across the test abnormal classes, and the direction depends on the test abnormal class' similarity to the train abnormal class.

We also observe that when using the reconstruction based models (AE, SAE, ABC), the performance for StatLog is much worse than the hypersphere based models. This result is in fact consistent with what has been reported in the literature—Ishii et al. (2020) reports a similar low performance of reconstruction based model on StatLog, which was trained on normal data (cf Table 4 of Ishii et al. (2020)). We consider this as a result potentially arising from specific latent spaces of data on the reconstruction based models, and leave improvement of these reconstruction models to future work.

It is worth highlighting that although these reconstruction based models demonstrate inferior performance on StatLog when training only on normal data, adding training abnormal set under Scenario 1 does demonstrate a similar behavior to that of the hypersphere based models. When training the SAE model with (biased) anomaly data, we handle the exploding loss issue in a similar way as Ruff et al. (2020b): For the anomaly class, we consider a loss function which takes the form of $\frac{1}{\text{reconstruction error}}$. We found that this design of loss function easily converges in practice with few loss explosion issues on reconstruction based models.

training normal = `top`, training abnormal = `shirt`

| Test data | Deep SVDD | Deep SAD | HSC | AE | SAE | ABC | $L^2$ to `shirt` |
|---|---|---|---|---|---|---|---|
| shirt | $0.09 \pm 0.01$ | $0.71 \pm 0.01$ ↑ | $0.70 \pm 0.01$ ↑ | $0.12 \pm 0.01$ | $0.72 \pm 0.01$ ↑ | $0.72 \pm 0.01$ ↑ | 0 |
| pullover | $0.13 \pm 0.02$ | $0.90 \pm 0.01$ ↑ | $0.89 \pm 0.01$ ↑ | $0.19 \pm 0.02$ | $0.84 \pm 0.02$ ↑ | $0.85 \pm 0.01$ ↑ | 0.01 |
| coat | $0.14 \pm 0.03$ | $0.92 \pm 0.02$ ↑ | $0.92 \pm 0.01$ ↑ | $0.15 \pm 0.02$ | $0.92 \pm 0.02$ ↑ | $0.92 \pm 0.01$ ↑ | 0.01 |
| dress | $0.17 \pm 0.03$ | $0.24 \pm 0.03$ ↑ | $0.24 \pm 0.03$ ↑ | $0.11 \pm 0.01$ | $0.20 \pm 0.03$ ↑ | $0.21 \pm 0.03$ ↑ | 0.04 |
| bag | $0.49 \pm 0.07$ | $0.38 \pm 0.08$ ↓ | $0.36 \pm 0.07$ ↓ | $0.70 \pm 0.04$ | $0.52 \pm 0.09$ ↓ | $0.53 \pm 0.07$ ↓ | 0.04 |
| trouser | $0.32 \pm 0.10$ | $0.07 \pm 0.04$ ↓ | $0.06 \pm 0.03$ ↓ | $0.59 \pm 0.04$ | $0.07 \pm 0.04$ ↓ | $0.16 \pm 0.07$ ↓ | 0.06 |
| boot | $0.92 \pm 0.03$ | $0.29 \pm 0.15$ ↓ | $0.27 \pm 0.16$ ↓ | $0.98 \pm 0.02$ | $0.90 \pm 0.09$ ↓ | $0.90 \pm 0.08$ ↓ | 0.08 |
| sandal | $0.30 \pm 0.04$ | $0.26 \pm 0.08$ ↓ | $0.26 \pm 0.12$ ↓ | $0.82 \pm 0.02$ | $0.46 \pm 0.10$ ↓ | $0.56 \pm 0.09$ ↓ | 0.09 |
| sneaker | $0.55 \pm 0.09$ | $0.12 \pm 0.10$ ↓ | $0.14 \pm 0.12$ ↓ | $0.74 \pm 0.09$ | $0.47 \pm 0.19$ ↓ | $0.46 \pm 0.18$ ↓ | 0.10 |

Table 2: The model TPR under scenario 1, Fashion-MNIST. The normal class `top` is similar to the abnormal training class `shirt`. Their $L^2$ distance = 0.02.

training normal = `very damp grey soil`, training abnormal = `damp grey soil`

| Test data | Deep SVDD | Deep SAD | HSC | AE | SAE | ABC | $L^2$ to `damp grey soil` |
|---|---|---|---|---|---|---|---|
| damp grey soil | $0.12 \pm 0.05$ | $0.81 \pm 0.02$ ↑ | $0.80 \pm 0.02$ ↑ | $0.00 \pm 0.00$ | $0.08 \pm 0.02$ ↑ | $0.01 \pm 0.01$ ↓ | 0 |
| red soil | $0.67 \pm 0.16$ | $0.92 \pm 0.05$ ↑ | $0.91 \pm 0.05$ ↑ | $0.00 \pm 0.00$ | $0.00 \pm 0.00$ | $0.00 \pm 0.00$ | 4.39 |
| grey soil | $0.45 \pm 0.17$ | $0.92 \pm 0.02$ ↑ | $0.92 \pm 0.02$ ↑ | $0.01 \pm 0.00$ | $0.04 \pm 0.02$ ↑ | $0.02 \pm 0.02$ | 4.42 |
| vegetable soil | $0.40 \pm 0.10$ | $0.19 \pm 0.04$ ↓ | $0.18 \pm 0.04$ ↓ | $0.35 \pm 0.01$ | $0.35 \pm 0.01$ | $0.35 \pm 0.00$ | 5.44 |
| cotton crop | $0.96 \pm 0.06$ | $0.80 \pm 0.06$ ↓ | $0.70 \pm 0.08$ ↓ | $0.89 \pm 0.02$ | $0.90 \pm 0.01$ | $0.90 \pm 0.01$ | 11.46 |

Table 3: The model TPR under scenario 1, StatLog. The normal class `very damp grey soil` is similar to the abnormal training class `damp grey soil`. Their $L^2$ distance = 3.63.

training normal = `normal`, training abnormal = `NB-10ms`

| Test data | Deep SVDD | Deep SAD | HSC | AE | SAE | ABC | $L^2$ to `NB-10ms` |
|---|---|---|---|---|---|---|---|
| NB-10ms | $0.03 \pm 0.01$ | $0.99 \pm 0.02$ ↑ | $0.99 \pm 0.01$ ↑ | $0.02 \pm 0.01$ | $0.97 \pm 0.05$ ↑ | $0.92 \pm 0.03$ ↑ | 0 |
| NB-5ms | $0.02 \pm 0.00$ | $0.93 \pm 0.05$ ↑ | $0.82 \pm 0.07$ ↑ | $0.00 \pm 0.00$ | $0.96 \pm 0.02$ ↑ | $0.99 \pm 0.01$ ↑ | 6.22 |
| WB-nlos | $0.99 \pm 0.00$ | $0.38 \pm 0.08$ ↓ | $0.43 \pm 0.10$ ↓ | $0.89 \pm 0.03$ | $0.54 \pm 0.02$ ↓ | $0.47 \pm 0.02$ ↓ | 32.40 |
| WB-los | $0.99 \pm 0.00$ | $0.50 \pm 0.10$ ↓ | $0.53 \pm 0.15$ ↓ | $0.92 \pm 0.03$ | $0.51 \pm 0.07$ ↓ | $0.50 \pm 0.03$ ↓ | 43.01 |

Table 4: The model TPR under scenario 1, Cellular Spectrum Misuse. The normal class is similar to the abnormal training class `NB-10ms`, and the $L^2$ distance between the two is 6.17.

**Scenario 2.** We consider scenario 2 where the training normal set is visually dissimilar to the training abnormal set. The detailed TPR result of all six models, and three real-life datasets are in Table 5 – 7. Like the above, in each table, the abnormal test classes are sorted by decreasing similarity to the abnormal training class. Like the above, ↑ indicates that the supervised model has a higher recall than the semi-supervised model; ↓ indicates the other direction.

Different from Scenario 1, here we observe mostly upward changes. Again we observe poorer performance of AE, SAE, ABC on StatLog compared to the hypersphere-based models.

training normal = `top`, training abnormal = `sneaker`

| Test data | Deep SVDD | Deep SAD | HSC | AE | SAE | ABC | $L^2$ to `sneaker` |
|---|---|---|---|---|---|---|---|
| sneaker | $0.55 \pm 0.09$ | $1.00 \pm 0.00$ ↑ | $1.00 \pm 0.00$ ↑ | $0.74 \pm 0.09$ | $1.00 \pm 0.00$ ↑ | $1.00 \pm 0.00$ ↑ | 0 |
| sandal | $0.30 \pm 0.04$ | $0.99 \pm 0.01$ ↑ | $0.98 \pm 0.02$ ↑ | $0.82 \pm 0.02$ | $1.00 \pm 0.00$ ↑ | $1.00 \pm 0.00$ ↑ | 0.02 |
| boot | $0.92 \pm 0.03$ | $1.00 \pm 0.00$ ↑ | $0.97 \pm 0.02$ ↑ | $0.98 \pm 0.02$ | $1.00 \pm 0.00$ ↑ | $1.00 \pm 0.00$ ↑ | 0.07 |
| bag | $0.49 \pm 0.07$ | $0.80 \pm 0.05$ ↑ | $0.81 \pm 0.11$ ↑ | $0.70 \pm 0.03$ | $0.84 \pm 0.03$ ↑ | $0.82 \pm 0.03$ ↑ | 0.07 |
| shirt | $0.09 \pm 0.01$ | $0.11 \pm 0.02$ ↑ | $0.12 \pm 0.01$ ↑ | $0.12 \pm 0.01$ | $0.13 \pm 0.01$ ↑ | $0.15 \pm 0.01$ ↑ | 0.10 |
| trouser | $0.32 \pm 0.09$ | $0.31 \pm 0.10$ | $0.11 \pm 0.12$ ↓ | $0.58 \pm 0.04$ | $0.58 \pm 0.03$ | $0.58 \pm 0.05$ | 0.12 |
| dress | $0.16 \pm 0.03$ | $0.16 \pm 0.04$ | $0.11 \pm 0.01$ ↓ | $0.11 \pm 0.01$ | $0.11 \pm 0.01$ | $0.12 \pm 0.01$ | 0.13 |
| pullover | $0.13 \pm 0.02$ | $0.13 \pm 0.03$ | $0.14 \pm 0.05$ | $0.19 \pm 0.02$ | $0.21 \pm 0.03$ | $0.19 \pm 0.02$ | 0.13 |
| coat | $0.14 \pm 0.03$ | $0.13 \pm 0.03$ | $0.13 \pm 0.06$ | $0.15 \pm 0.02$ | $0.16 \pm 0.02$ | $0.15 \pm 0.02$ | 0.14 |

Table 5: The model TPR under scenario 2, Fashion-MNIST. The normal class `top` is dissimilar to the abnormal training class `sneaker`, and the $L^2$ distance between the two is 0.13.

training normal = `very damp grey soil`, training abnormal = `red soil`

| Test data | Deep SVDD | Deep SAD | HSC | AE | SAE | ABC | $L^2$ to `red soil` |
|---|---|---|---|---|---|---|---|
| red soil | $0.69 \pm 0.12$ | $1.00 \pm 0.00$ ↑ | $1.00 \pm 0.00$ ↑ | $0.00 \pm 0.00$ | $0.21 \pm 0.01$ ↑ | $0.20 \pm 0.00$ ↑ | 0 |
| damp grey soil | $0.12 \pm 0.05$ | $0.25 \pm 0.04$ ↑ | $0.22 \pm 0.04$ ↑ | $0.00 \pm 0.00$ | $0.00 \pm 0.00$ | $0.00 \pm 0.00$ | 4.39 |
| grey soil | $0.43 \pm 0.16$ | $0.68 \pm 0.12$ ↑ | $0.53 \pm 0.09$ ↑ | $0.01 \pm 0.00$ | $0.02 \pm 0.00$ | $0.01 \pm 0.00$ | 5.48 |
| vegetable soil | $0.40 \pm 0.10$ | $0.76 \pm 0.07$ ↑ | $0.76 \pm 0.07$ ↑ | $0.35 \pm 0.00$ | $0.42 \pm 0.02$ ↑ | $0.42 \pm 0.02$ ↑ | 6.63 |
| cotton crop | $0.96 \pm 0.06$ | $1.00 \pm 0.00$ ↑ | $1.00 \pm 0.00$ ↑ | $0.89 \pm 0.00$ | $0.93 \pm 0.01$ ↑ | $0.93 \pm 0.01$ ↑ | 8.97 |

Table 6: The model TPR under scenario 2, StatLog. The normal class `very damp grey soil` is dissimilar to the abnormal training class `red soil`, and the $L^2$ distance between the two is 8.48.

training normal = `normal`, training abnormal = `WB-los`

| Test data | Deep SVDD | Deep SAD | HSC | AE | SAE | ABC | $L^2$ to `WB-los` |
|---|---|---|---|---|---|---|---|
| WB-los | $0.99 \pm 0.00$ | $1.00 \pm 0.01$ ↑ | $1.00 \pm 0.01$ ↑ | $0.92 \pm 0.03$ | $0.95 \pm 0.04$ ↑ | $1.00 \pm 0.02$ ↑ | 0 |
| WB-nlos | $0.99 \pm 0.00$ | $1.00 \pm 0.01$ ↑ | $1.00 \pm 0.00$ ↑ | $0.89 \pm 0.03$ | $0.94 \pm 0.03$ ↑ | $0.96 \pm 0.02$ ↑ | 14.39 |
| NB-10ms | $0.03 \pm 0.01$ | $0.06 \pm 0.01$ ↑ | $0.05 \pm 0.02$ | $0.02 \pm 0.01$ | $0.03 \pm 0.00$ ↑ | $0.04 \pm 0.01$ | 43.01 |
| NB-5ms | $0.02 \pm 0.00$ | $0.03 \pm 0.01$ | $0.02 \pm 0.00$ | $0.00 \pm 0.00$ | $0.02 \pm 0.00$ ↑ | $0.02 \pm 0.01$ ↑ | 44.37 |

Table 7: The model TPR under scenario 2, Cellular Spectrum Misuse. The normal class is dissimilar to the abnormal training class `WB-los`, and the $L^2$ distance between the two is 43.84.

**Scenario 3.** We run three configurations of grouped abnormal training on Fashion-MNIST (training normal: `top`; training abnormal: `shirt` & `sneaker`) by varying the weights of the two abnormal classes in training (0.5/0.5, 0.9/0.1, 0.1/0.9). Again ↑ indicates that the supervised model has a higher recall than the semi-supervised model; ↓ indicates the other direction. Under these settings, we observe downward bias (↓) for one abnormal test class `trouser` and upward bias for most other classes.

training normal = `top`, training abnormal = 50% `shirt` and 50% `sneaker`

| Test data | Deep SVDD | Deep SAD | HSC | AE | SAE | ABC | $L^2$ to `shirt` | $L^2$ to `sneaker` |
|---|---|---|---|---|---|---|---|---|
| `shirt` | $0.09 \pm 0.01$ | $0.69 \pm 0.01$ ↑ | $0.69 \pm 0.02$ ↑ | $0.12 \pm 0.01$ | $0.67 \pm 0.01$ ↑ | $0.66 \pm 0.01$ ↑ | 0 | 0.10 |
| `sneaker` | $0.55 \pm 0.09$ | $1.00 \pm 0.00$ ↑ | $1.00 \pm 0.00$ ↑ | $0.74 \pm 0.09$ | $1.00 \pm 0.00$ ↑ | $1.00 \pm 0.00$ ↑ | 0.10 | 0 |
| `pullover` | $0.13 \pm 0.02$ | $0.90 \pm 0.01$ ↑ | $0.90 \pm 0.01$ ↑ | $0.19 \pm 0.02$ | $0.82 \pm 0.02$ ↑ | $0.83 \pm 0.02$ ↑ | 0.01 | 0.13 |
| `coat` | $0.14 \pm 0.03$ | $0.91 \pm 0.02$ ↑ | $0.90 \pm 0.01$ ↑ | $0.15 \pm 0.02$ | $0.86 \pm 0.02$ ↑ | $0.87 \pm 0.02$ ↑ | 0.01 | 0.14 |
| `dress` | $0.17 \pm 0.03$ | $0.23 \pm 0.04$ ↑ | $0.24 \pm 0.04$ ↑ | $0.11 \pm 0.01$ | $0.19 \pm 0.03$ ↑ | $0.18 \pm 0.02$ ↑ | 0.04 | 0.13 |
| `bag` | $0.49 \pm 0.07$ | $0.63 \pm 0.06$ ↑ | $0.62 \pm 0.07$ ↑ | $0.70 \pm 0.03$ | $0.76 \pm 0.05$ ↑ | $0.78 \pm 0.03$ ↑ | 0.04 | 0.07 |
| `trouser` | $0.32 \pm 0.10$ | $0.05 \pm 0.04$ ↓ | $0.04 \pm 0.02$ ↓ | $0.59 \pm 0.04$ | $0.22 \pm 0.08$ ↓ | $0.34 \pm 0.06$ ↓ | 0.06 | 0.12 |
| `boot` | $0.92 \pm 0.03$ | $0.95 \pm 0.03$ | $0.95 \pm 0.03$ | $0.98 \pm 0.02$ | $1.00 \pm 0.00$ ↑ | $1.00 \pm 0.00$ ↑ | 0.08 | 0.07 |
| `sandal` | $0.30 \pm 0.04$ | $0.92 \pm 0.04$ ↑ | $0.92 \pm 0.04$ ↑ | $0.82 \pm 0.02$ | $0.96 \pm 0.01$ ↑ | $0.97 \pm 0.01$ ↑ | 0.09 | 0.02 |

Table 8: The model TPR under configuration 1 of weighted mixture training on Fashion-MNIST.

training normal = `top`, training abnormal = 90% `shirt` and 10% `sneaker`

| Test data | Deep SVDD | Deep SAD | HSC | AE | SAE | ABC | $L^2$ to `shirt` | $L^2$ to `sneaker` |
|---|---|---|---|---|---|---|---|---|
| `shirt` | $0.09 \pm 0.01$ | $0.70 \pm 0.01$ ↑ | $0.70 \pm 0.01$ ↑ | $0.12 \pm 0.01$ | $0.72 \pm 0.01$ ↑ | $0.71 \pm 0.01$ ↑ | 0 | 0.10 |
| `sneaker` | $0.55 \pm 0.09$ | $1.00 \pm 0.00$ ↑ | $1.00 \pm 0.00$ ↑ | $0.74 \pm 0.09$ | $1.00 \pm 0.00$ ↑ | $1.00 \pm 0.00$ ↑ | 0.10 | 0 |
| `pullover` | $0.13 \pm 0.02$ | $0.90 \pm 0.01$ ↑ | $0.89 \pm 0.01$ ↑ | $0.19 \pm 0.02$ | $0.84 \pm 0.02$ ↑ | $0.84 \pm 0.02$ ↑ | 0.01 | 0.13 |
| `coat` | $0.14 \pm 0.03$ | $0.91 \pm 0.02$ ↑ | $0.91 \pm 0.02$ ↑ | $0.15 \pm 0.02$ | $0.91 \pm 0.02$ ↑ | $0.90 \pm 0.02$ ↑ | 0.01 | 0.14 |
| `dress` | $0.17 \pm 0.03$ | $0.23 \pm 0.03$ ↑ | $0.24 \pm 0.03$ ↑ | $0.11 \pm 0.01$ | $0.19 \pm 0.03$ ↑ | $0.20 \pm 0.03$ ↑ | 0.04 | 0.13 |
| `bag` | $0.49 \pm 0.07$ | $0.56 \pm 0.08$ | $0.57 \pm 0.07$ | $0.70 \pm 0.03$ | $0.67 \pm 0.06$ | $0.68 \pm 0.05$ | 0.04 | 0.07 |
| `trouser` | $0.32 \pm 0.10$ | $0.06 \pm 0.04$ ↓ | $0.06 \pm 0.03$ ↓ | $0.59 \pm 0.04$ | $0.10 \pm 0.06$ ↓ | $0.20 \pm 0.08$ ↓ | 0.06 | 0.12 |
| `boot` | $0.92 \pm 0.03$ | $0.87 \pm 0.08$ | $0.88 \pm 0.05$ | $0.98 \pm 0.02$ | $0.99 \pm 0.01$ | $0.99 \pm 0.00$ | 0.08 | 0.07 |
| `sandal` | $0.30 \pm 0.04$ | $0.84 \pm 0.06$ ↑ | $0.83 \pm 0.05$ ↑ | $0.82 \pm 0.02$ | $0.90 \pm 0.02$ ↑ | $0.94 \pm 0.02$ ↑ | 0.09 | 0.02 |

Table 9: The model TPR under configuration 2 of weighted mixture training on Fashion-MNIST.

training normal = `top`, training abnormal = 10% `shirt` and 90% `sneaker`

| Test data | Deep SVDD | Deep SAD | HSC | AE | SAE | ABC | $L^2$ to `shirt` | $L^2$ to `sneaker` |
|---|---|---|---|---|---|---|---|---|
| `shirt` | $0.09 \pm 0.01$ | $0.61 \pm 0.02$ ↑ | $0.60 \pm 0.02$ ↑ | $0.12 \pm 0.01$ | $0.54 \pm 0.02$ ↑ | $0.54 \pm 0.01$ ↑ | 0 | 0.10 |
| `sneaker` | $0.55 \pm 0.09$ | $1.00 \pm 0.00$ ↑ | $1.00 \pm 0.00$ ↑ | $0.74 \pm 0.09$ | $1.00 \pm 0.00$ ↑ | $1.00 \pm 0.00$ ↑ | 0.10 | 0 |
| `pullover` | $0.13 \pm 0.02$ | $0.85 \pm 0.02$ ↑ | $0.84 \pm 0.02$ ↑ | $0.19 \pm 0.02$ | $0.74 \pm 0.03$ ↑ | $0.74 \pm 0.03$ ↑ | 0.01 | 0.13 |
| `coat` | $0.14 \pm 0.03$ | $0.79 \pm 0.03$ ↑ | $0.77 \pm 0.03$ ↑ | $0.15 \pm 0.02$ | $0.67 \pm 0.04$ ↑ | $0.68 \pm 0.03$ ↑ | 0.01 | 0.14 |
| `dress` | $0.17 \pm 0.03$ | $0.13 \pm 0.03$ ↓ | $0.11 \pm 0.03$ ↓ | $0.11 \pm 0.01$ | $0.12 \pm 0.01$ | $0.11 \pm 0.01$ | 0.04 | 0.13 |
| `bag` | $0.49 \pm 0.07$ | $0.82 \pm 0.05$ ↑ | $0.84 \pm 0.05$ ↑ | $0.70 \pm 0.03$ | $0.85 \pm 0.03$ ↑ | $0.86 \pm 0.02$ ↑ | 0.04 | 0.07 |
| `trouser` | $0.32 \pm 0.10$ | $0.10 \pm 0.08$ ↓ | $0.05 \pm 0.05$ ↓ | $0.59 \pm 0.04$ | $0.45 \pm 0.07$ ↓ | $0.50 \pm 0.05$ ↓ | 0.06 | 0.12 |
| `boot` | $0.92 \pm 0.03$ | $1.00 \pm 0.00$ ↑ | $0.99 \pm 0.01$ ↑ | $0.98 \pm 0.02$ | $0.98 \pm 0.00$ | $1.00 \pm 0.00$ | 0.08 | 0.07 |
| `sandal` | $0.30 \pm 0.04$ | $0.99 \pm 0.01$ ↑ | $0.83 \pm 0.01$ ↑ | $0.82 \pm 0.02$ | $0.98 \pm 0.00$ ↑ | $0.99 \pm 0.00$ ↑ | 0.09 | 0.02 |

Table 10: The model TPR under configuration 3 of weighted mixture training on Fashion-MNIST.

