# OpenReview forum: "Understanding the Effect of Bias in Deep Anomaly Detection"
_ICLR.cc/2021/Conference — Reject_

### Official Review · AnonReviewer4 · 2020-10-22
**Important investigation of the effect of biased anomaly datasets in semi-supervised anomaly detection. A more deep analysis of the theoretical and experimental results would improve the quality of this work.**

**Rating:** 6
**Confidence:** 2

**Review:**

======================
Additional reviews
======================
The authors have resolved some concerns, especially the explanation/justification of the experimental results in section 5.
As I have commented in the initial review, this paper provides suggestive experimental results (the effects of training anomaly dataset) for future anomaly detection research.
Therefore, I have decided to raise my rating from 5 to 6.
======================

Summary:
- This paper investigates the effects of biased training anomaly datasets on anomaly detection problems.
Specifically, this paper proposes the relative scoring bias, which is the difference of TPRs of two anomaly detectors when the FPR is below a certain value, to model the effects of a biased training anomaly dataset.
In addition, this paper also presents the finite sample rates for estimating the relative scoring bias.
This paper empirically analyzes the effect of a biased training anomaly dataset on the detection performance.

Pros:
- Investigating the effect of training anomaly datasets on anomaly detection is important and useful in anomaly detection studies.
- This paper establishes the finite sample rates for estimating the relative scoring bias for semi-supervised anomaly detection.
- Empirical evaluation results (especially, the results in Section 5) are interesting.

Cons:
- Evaluation for the convergence of the relative scoring bias/FPRs was not conducted with the real-world datasets.

Detailed comments and questions:
- As the paper mentioned, semi-supervised anomaly detection, which uses labeled anomaly data as well as normal/unlabeled data for training, has become a promising approach to improve the performance of anomaly detectors.
Since all types of anomaly data are difficult to collect due to the rarity of anomalies, it is typical to use biased anomaly datasets for learning the semi-supervised anomaly detectors.
Therefore, analyzing how/when the biased training anomaly dataset affects performance is important/useful in this research area.

- Although the results in Section 5 are really interesting/useful, can the authors explain why the phenomenon of results 5 occurs? That is, why performance on test data dissimilar to the training anomaly data (do not) becomes worse when anomaly and normal training data are similar (dissimilar)?

- The paper only evaluated the convergence of the relative scoring bias/FPRs on the simple synthetic dataset. Can the authors investigate it using the real-world datasets to empirically validate the Theorem 3?

- Figure 2 is a bit difficult to see. In particular, it is difficult to compare the variation of $\xi$ at $\alpha=0.2$ between $\alpha=0.01$ when $n=10000$.

Reasons for Score:
- To validate the theoretical results empirically, I would like to see the experimental results about the convergence of the relative scoring bias/FPRs with real-world datasets.
Although my current rating is 5, I would like to raise my score when the above questions are resolved.
Since I'm not familiar with the PAC framework, I cannot judge the novelty of theoretical results (Theorem 3) well. Therefore, my confidence is 2.

Minor commetns:
- There are some typos:
 - p4. Algorithm 1: FPR (recall) -> TPR (recall)
 - below Eq. (3.5) in p4. $s(x_j) \leq \tau $ ->  $s(x_j) \geq \tau $
 - p8. Fashion-MNISST -> Fashion-MNIST

---

> ### Author Response · Authors · 2020-11-16
> **Authors' response to AnonReviewer4 (Clarification to questions in experimental details)**
>
> Thank you for the detailed review and suggestions! Below please find our response (**A**) to the main questions (**Q1-3**).
>
> **Q1: [Justification and explanation of the empirical results in Section 5]**
> (“why performance on test data dissimilar to the training anomaly data (do not) becomes worse when anomaly and normal training data are similar (dissimilar)”)
>
> **A**: As pointed out by the reviewer, we empirically evaluated two biased training scenarios using multiple datasets:
> * Scenario 1: the training anomaly and normal data are similar
> \
> We observed both downward and upward bias across anomaly classes, when comparing unsupervised and semi-supervised models.
> * Scenario 2: the training anomaly and normal data are dissimilar
> \
> We observed little downward change of performance.
>
> Our interpretation of the difference between two scenarios is as follows:
>
> * when training anomaly and normal data are similar,  adding new training anomaly will lead to a significant change in the training distribution, and hence have a large effect on the scoring function;
> * when the training anomaly and normal are dissimilar,  the change of distribution has less effect on  the deep anomaly detection models studied in this work -- because these dissimilar anomaly data are already deemed as anomaly by the unsupervised version, and the anomaly training further confirms such decision (rather than changing it).
>
> We plan to include this discussion in the revision. Does this address the comment?
>
> **Q2: [Justification of the theoretical results in Section 4 on real-world datasets]**
> (“Can the authors investigate it using the real-world datasets to empirically validate the Theorem 3?”)
>
> **A**:  Thank you for this excellent suggestion.  We are running experiments on benchmark datasets (e.g. ODDS) and the FashionMNIST dataset, to validate our theoretical results in Theorem 3, particularly the convergence result.
>
> From our initial results, we observe a consistent convergence behavior of the relative scoring bias on all the datasets, which aligns with the results in Figure 2, 8, 9, 10 that were obtained on simulated data. The current visualization of convergence results, which we will integrate into the revision, can be previewed at this anonymized link: https://gofile.io/d/hHk5HC. Below is a preview for the convergence of ξ on StatLog by ABC model (experiment details are in the file at the link), where we report ξ in the fashion of mean ± std.
>
> |          |  alpha=0.05&nbsp;&nbsp;&nbsp;&nbsp;&nbsp;&nbsp;   |   alpha=0.1&nbsp;&nbsp;&nbsp;&nbsp; &nbsp;&nbsp;  |   alpha=0.2   |
> | -------  | ------------  | ------------  | ------------  |
> | n = 100  | 0.116 ± 0.083 | 0.079 ± 0.060 | 0.074 ± 0.036 |
> | n = 200  | 0.114 ± 0.072 | 0.073 ± 0.047 | 0.071 ± 0.027 |
> | n = 500  | 0.120 ± 0.056 | 0.078 ± 0.037 | 0.076 ± 0.020 |
> | n = 1000 | 0.146 ± 0.014 | 0.098 ± 0.007 | 0.086 ± 0.004 |
>
> **Q3: [Clarification on plots]**
> (“Figure 2 is a bit difficult to see. In particular, it is difficult to compare the variation of ξ at α=0.2 between α=0.01 when n=10000”)
>
> **A**: Thanks for pointing this out. The empirical value of ξ converges as n grows. We will refine Figure 2’s visualization and will add the variance values to the description of the plots.
>
> Below is a preview of the description for Figure 2(a), where we report ξ in the fashion of mean ± std.
>
> |            |   alpha=0.01&nbsp;&nbsp;&nbsp;&nbsp;&nbsp;&nbsp;  |    alpha=0.05&nbsp;&nbsp;&nbsp;&nbsp;&nbsp;&nbsp;  |    alpha=0.1&nbsp;&nbsp;&nbsp;&nbsp;&nbsp;&nbsp;  |     alpha=0.2 |
> | ---------  | ------------  | -------------  | ------------  | ------------  |
> | n = 100    | 0.549 ± 0.093 | 0.556 ± 0.086  | 0.537 ± 0.090 | 0.538 ± 0.090 |
> | n = 1000   | 0.537 ± 0.038 | 0.553 ± 0.029  | 0.533 ± 0.033 | 0.534 ± 0.028 |
> | n = 10000  | 0.540 ± 0.015 | 0.554 ± 0.010  | 0.531 ± 0.010 | 0.533 ± 0.009 |

---

> > ### Comment · AnonReviewer4 · 2020-11-23
> > **Thank for your response**
> >
> > Thank you for your detailed response. The response has resolved my questions.
> > This discussion of the empirical results in section 5 would improve the paper's quality.
> > After leading other reviews/responses as well as mine, I think that this paper has sufficient contributions to ICLR 2021.
> > Therefore, I'm happy to raise my score from 5 to 6.

---

### Official Review · AnonReviewer1 · 2020-10-28
**Review AnonReviewer1**

**Rating:** 7
**Confidence:** 4

**Review:**


**UPDATE**

I acknowledge that I have read the author responses as well as the other reviews. I appreciate the clarifications and improvements made during the rebuttal phase, which I think have further strengthened this work.

I find the key contributions of this work to be (i) demonstrating that recent methods that include labeled anomalies into training can suffer from unfavorable biases, and (ii) providing a framework for a theoretical analysis of this setting.

Though I see that the presented results are somewhat what one would expect, to my knowledge such an analysis hasn't been carried out in the existing literature.

Since weak forms of supervision (here few labeled anomalies) appears to be a promising research direction for anomaly detection, I find this critical and rigorous analysis to be worth circulating the community.

For these reasons, I would keep my recommendation to accept this work (score: 7)

#####


**Summary**

This paper studies biases in semi-supervised anomaly detection which is the setting where in addition to mostly nominal data a few labeled anomalies are also available for training. A theoretical framework for the semi-supervised setting is introduced that is based on binary classification which formulates the objective of a scorer as seeking to maximize the detection recall (true positive rate (TPR)) at a given target false positive rate (FPR). Using this framework, a relative scoring bias is derived that enables to assess the relative performance difference between unsupervised and semi-supervised detectors. Furthermore, finite sample rates are derived for this relative scoring bias, which subsequently are also validated empirically via synthetic simulations. Finally, an empirical evaluation that includes six recent state-of-the-art deep anomaly detection methods (Deep SVDD, Deep SAD, HSC, AE, SAE, and ABC) is presented on Fashion-MNIST, Statlog (Landsat Satellite), and ImageNet that demonstrates and highlights scenarios where the bias of a labeled (unrepresentative) anomaly set can be useful, but also harmful for anomaly detection performance.


**Pros**
+ The paper presents a novel theoretical PAC framework for analyzing and understanding bias in semi-supervised anomaly detection. The framework extends a previous classification-based view on anomaly detection [4] to the semi-supervised setting.
+ Deep semi-supervised anomaly detection methods [2, 5, 7] that aim to include and learn from labeled anomalies is a timely topic of high practical relevance.
+ The experimental evaluation demonstrates that including labeled anomalies might introduce an unfavorable bias that can decrease detection performance, which is an important insight.
+ The paper has a clear structure and is easy to follow.

**Cons**
- Some related work is missing, especially previous classification-based views on anomaly detection [8].
- There are some questions left open (see below).
- The current manuscript includes some (minor) typos that should be fixed.


**Recommendation**

I recommend to accept this paper.

The paper presents a well-motivated and useful theoretical framework for the timely and relevant semi-supervised anomaly detection setting. The arguments and derivations are technically correct. To my knowledge, this is also the first instance of a finite sample complexity bound on the scoring bias for this setting. The theoretical claims are validated through simulations and tested on real-world datasets in a scientifically rigorous manner. An important message of the analysis is that including labeled anomalies can introduce a bias that can be harmful for anomaly detection performance. In this regard, I think the paper also covers important ground for future analysis and towards building semi-supervised models that are unbiased.


**Questions**

(1) How does the presented view compare to well-known previous classification-based views [8]?

(2) ‘for $\xi$, it also converges to a certain level.’ Specifically the level predicted by the bound of Theorem 3?

(3) How did you stabilize maximizing the reconstruction error for labeled anomalies in SAE? I suspect optimizing this objective is
unstable and prone to blow up.

(4) Scenario 2 would make a compelling case for using Outlier Exposure [1]. Did you conduct such experiments similar to [6]?

(5) At the end of Section 3, the empirical TPR estimate should have $s(x_j) > \tau$ in the indicator function, correct?

(6) In the infinite sample case in Section 4.1, do you refer to the Glivenko–Cantelli theorem when citing Parzen (1980)?


**Additional feedback and ideas for improvement**
- Include Outlier Exposure [1] in the experimental analysis.
- There exists further recent related work on biases in anomaly detection that observes that detectors may correctly detect anomalies, but based on wrong (spurious) features [3]. This should be added to the list of works studying biases in anomaly detection.


**Minor Comments**
1. The last paragraph in the Introduction, in which the contributions are listed, is a bit repetitive after the preceding paragraphs.
2. In Section 4.1, $F_0(t)$ and $F_a(t)$ should have $s(x) \leq t$ in their definition to be consistent, right?
3. The notation for the number of anomalous training samples mixes $m$ and $n_1$.
4. Proposition 1: ‘[...], the relative scoring bias *is* [...]’
5. In Section 4.1, after Proposition 1, the $\text{TPR}(s', \tau')$ function is missing parentheses.
6. Corollary 2: ‘Let $q$ be a fixed target FPR. [...] Then, the relative scoring bias is [...]’
7. Note that $\Phi$ denotes the cdf of the standard Gaussian.
8. Finite sample case: ‘[...], where we follow the convention to assume *that the anomaly data* amounts to [...]’
9. Theorem 3: The ‘-’ in the cdf superscripts should be ‘-1’. 10. There are spaces missing after ‘i.i.d.’ in the text.


#####

**References**

[1] D. Hendrycks, M. Mazeika, and T. G. Dietterich. Deep anomaly detection with outlier exposure. In ICLR, 2019.

[2] D. Hendrycks, M. Mazeika, S. Kadavath, and D. Song. Using self-supervised learning can improve model robustness and uncertainty. In NeurIPS, pages 15637–15648, 2019.

[3] J. Kauffmann, L. Ruff, G. Montavon, and K.-R. Müller. The Clever Hans effect in anomaly detection. arXiv preprint arXiv:2006.10609, 2020.

[4] S. Liu, R. Garrepalli, T. Dietterich, A. Fern, and D. Hendrycks. Open category detection with PAC guarantees. In ICML, volume 80, pages 3169–3178, 2018.

[5] G. Pang, C. Shen, and A. van den Hengel. Deep anomaly detection with deviation networks. In KDD, pages 353–362, 2019.

[6] L. Ruff, R. A. Vandermeulen, B. J. Franks, K.-R. Müller, and M. Kloft. Rethinking assumptions in deep anomaly detection. arXiv preprint arXiv:2006.00339, 2020.

[7] L. Ruff, R. A. Vandermeulen, N. Görnitz, A. Binder, E. Müller, K.-R. Müller, and M. Kloft. Deep semi-supervised anomaly detection. In ICLR, 2020.

[8] I. Steinwart, D. Hush, and C. Scovel. A classification framework for anomaly detection. Journal of Machine Learning Research, 6(Feb):211–232, 2005.

---

> ### Author Response · Authors · 2020-11-16
> **Authors' response to AnonReviewer1**
>
> Thank you for the detailed review and insightful comments. Below please find our response (**A**) to the main questions (**Q1-7**).
>
> **Q1: [Positioning of the work]**
> (“How does the presented view compare to well-known previous classification-based views [8]”)
>
> **A**: The view presented in our work differs from the classification-based view (Steinwart et al. (2005), i.e. [8]), mainly in the formulation of the objective and performance measure. Steinwart et al. (2005) consider anomaly detection as a density level set detection problem. Their performance measure is essentially equivalent to the density of the mis-predicted labels---in other words, FP + FN---and the goal is to minimize the mis-prediction risk.
>
> In comparison, our formulation (Eq. (1)) adopts a different, yet common performance measure considered in the recent literature, namely Recall (i.e., TPR= TP/(TP + FN)) subject to achieving a given False Positive Rate (where FPR=FP/(FP+TN)).
>
> The motivation of adopting this formulation is that it (arguably) better reflects the practical usage of AD algorithms: often, the detection threshold is first set to guarantee a preset false positive rate, and the goal is therefore to maximize the recall. This formulation aligns with the settings of many contemporary work in anomaly detection, e.g., Liu et al (2018) consider the open category detection problem which falls under a similar regime.
>
> We plan to add text to our revision to explicitly describe the difference between the two views. Thank you for bringing this up.
>
> **Q2: [Convergence of ξ]**
> (“the level predicted by the bound of Theorem 3”)
>
> **A**: Please note that our theory implies the rate of convergence of the relative scoring bias, but does not imply/predict the value of the asymptotic bias.
>
> **Q3: [Convergence of quantile functions]**
> (“In the infinite sample case in Section 4.1, do you refer to the Glivenko–Cantelli theorem when citing Parzen (1980)?”)
>
> **A**: Thanks for raising this clarification question. We refer to both the Skorokhod's representation theorem and Theorem 2A of Parzen (1980) (https://apps.dtic.mil/dtic/tr/fulltext/u2/a093000.pdf), which states the convergence of the empirical quantile function and the empirical distribution function (here we need both to establish Proposition 1). The Glivenko–Cantelli theorem, combined with Theorem 2A of Parzen (1980),  also suffice to give us the same result. We plan to include these explicit references to Parzen (1980) in our revision. We hope this addresses the comment.
>
> **Q4: [Experimental setup and details]**
> (“How did you stabilize maximizing the reconstruction error for labeled anomalies in SAE? I suspect optimizing this objective is unstable and prone to blow up.”)
>
> **A**: We have considered two versions of loss functions in SAE to handling exploding loss; both have been well explored in the literature:
> * Du et al., 2019, “Lifelong Anomaly Detection Through Unlearning”.
> \
> Du et al., 2019 proposed to minimize the reconstruction loss for the normal class, while minimizing an augmented loss function for the anomaly class which takes the form of a bounded negative reconstruction error.
>
> * Ruff et al., 2020, “Deep Semi-Supervised Anomaly Detection”.
> \
> Ruff et al., 2020 proposed to minimize the distance to a latent hypercenter for the normal class. For the anomaly class, they considered a different loss design which takes the form of (1 / distance to the latent hypercenter). In our context, we replace the distance to the latent hypercenter by the reconstruction error.
>
> We found that the latter converges better in practice with few loss explosion issues on reconstruction based models, thus we adopted it for our experiments. We will add text in the revision to clarify the experimental setup for SAE.
>
> **Q5: [Results on “outlier exposure”]**
> (“Scenario 2 would make a compelling case for using Outlier Exposure. Did you conduct such experiments similar to [6]?”)
>
> **A**: Yes, we have conducted these experiments.
>
> In the original work of Ruff et al. 2020, they introduced hypersphere classifier (HSC) as a representative of the Outlier Exposure approach to anomaly detection. In our experiments, we have evaluated HSC (e.g. Figure 8, Table 2-4) across multiple datasets.
>
> **Q6: [Additional related work]**
> (“[3] should be added to the list of works studying biases in anomaly detection.”)
>
> **A**: Thanks for providing the reference. We will include it in the related work section in the revision.
>
> **Q7: [Typos and other issues]**
> (“At the end of Section 3, the empirical TPR estimate should have s(xj)>τ in the indicator function, correct?”, and minor comments)
>
> **A**: Yes, the s(xj)<τ in the indicator function is a typo. Thanks for pointing this out, and we appreciate the detailed suggestions and comments on these editorial issues. We will incorporate all the comments in the revision.

---

### Official Review · AnonReviewer2 · 2020-10-29
**Good work but have major issues in the fundamental assumptions**

**Rating:** 4
**Confidence:** 5

**Review:**

This paper studies the potential bias in deep semi-supervised anomaly detection. The bias is evaluated in terms of TPR rate given a fixed FPR rate. It uses the anomaly scores output by unsupervised anomaly detectors as a benchmark to examine the relative scoring bias in deep semi-supervised anomaly detectors. It further studies the finite sample rate for this type of scoring bias. This type of bias is verified using some synthetic and real-world datasets. The empirical results also show the potential impact of this bias on several anomaly detectors.

Overall, the paper is well written and studies an important problem in anomaly detection. A number of theoretical and empirical results are presented to justify the arguments.

However, there are some major issues. Particularly, the three claimed contributions are weak and/or built upon some inappropriate foundations. The first contribution is something partially or fully demonstrated in several exiting work  (Pang et al. (2019); Daniel et al. (2019); Yamanaka et al. (2019); Ruff et al. (2020b;a)). It is a quite straightforward phenomenon. Using the labeled anomaly data to train and validate the trained model of course results in some sort of inductive bias. This is a fundamental assumption in supervised learning like classification. In anomaly detection, such bias is often desired in the sense that we want our detectors to identify anomalies similar to those labeled anomaly data. By contrast, there can be novel types of anomalies that can be very different from the known anomalies. Those novel anomalies cannot be detected if fitting only to the known anomalies. Therefore, this conclusion is straightforward. Further, as anomaly detection has this type of crucial difference compared to classification, the idea of maximizing the TPR rate given a fixed FPR rate using a small labeled anomaly data, or the formulation of semi-supervised anomaly detection as a general supervised learning problem,  is ill-posed. This lets me doubt the importance of this work.

I also have major concerns over the definition of relative scoring bias. Why is it reasonable to use a unsupervised anomaly detector to serve as an approximator of the optimal anomaly detector? It does not make much sense to me. However, this is the fundamental assumption of the whole work.

Some other issues. (1) The title is also inappropriate. It should explicitly limit the scope to deep anomaly detection. This is what this work is about. (2) There are inconsistent  results in terms of the trending in tables 2-4 and tables 5-7, such as Deep SAD. Why would this happen? Also, the performance of each detector on individual datasets varies significantly. This should also be a key factor to be carefully considered before drawing any conclusions. Additionally, what is the standard deviation of these results? if it is large, we may also need to consider the contribution of this factor in the 'bias'.

---

> ### Author Response · Authors · 2020-11-16
> **Authors' response to AnonReviewer2 (Clarifications to questions in contribution and significance of the work)**
>
> Thank you for the detailed review and comments. Below please find our response (A) to the questions regarding the contribution and significance of this work (Q1-2).
>
> **Q1: [Positioning of this work]**
> (“the first claimed contribution is demonstrated in several existing work”)
>
> **A**: Thank you for the comment. The focus of existing works (e.g., Pang et al. (2019); Daniel et al. (2019); Yamanaka et al. (2019); Ruff et al. (2020b;a)) is to develop semi-supervised anomaly detection algorithms that leverage available labeled anomalies. They show that training using these labeled anomalies will (always) improve anomaly detection accuracy.
>
> Our work differs from these works by considering a more general scenario where the labeled anomalies used for training are some subclass (A_i) of the entire collection of anomalies (A). In particular, our work seeks to understand how existing deep learning models, when trained on anomaly subclass A_i, will perform on A, with detailed results on each subclass of A. Along this line, we discovered the biasing effect from both limited samples and skewed training data (i.e. the presence of both positive and negative effects across classes), and its dependency on A_i. We believe this issue has not been formally or empirically studied by existing works.
>
> Your comment helps us realize that we should clarify our contribution and state the difference from these existing works, which we plan to do in the rebuttal revision. We hope that this modification addresses the comment.
>
>
> **Q2: [Significance of this work]**
> (“It is a quite straightforward phenomenon”, and “This (inductive bias) is a fundamental assumption in supervised learning like classification”)
>
> **A**:  We agree that at a high-level, the bias effect in anomaly detection is seemingly a natural extension from the supervised classification scenario. Yet there lacks a rigorous and systematic study of such phenomenon, both theoretically and empirically. Our work seeks to address this gap.
>
> Below we highlight the key distinctions and significance of our work, in comparison to existing work:
>
> At the theoretical front:
> * When there’s no sampling bias (i.e. training and test data both sampled i.i.d.), there has been extensive theoretical study in both the classification (e.g., Vapnik, 1984) and anomaly detection problems (e.g., Liu et al. 2018), often in terms of finite sample analysis for achieving a target performance.
> \
> On the other hand, when sampling bias is present, it has been extensively explored as the domain adaptation problem in the context of classification (this was also pointed out by AnonReviewer3).   To the best of our knowledge, there has been little or no work in formally studying its impact on anomaly detection. Thus, our work -- namely the finite sample analysis on the effect of bias due to insufficient samples -- is making an initial but important contribution in closing the gap.
>
> At the empirical front:
> * We demonstrate scenarios in which the biased anomaly training can be useful or harmful in detecting different subclasses of anomalies seen at the test time. This finding exposes an issue/challenge in building semi-supervised anomaly detection models. As we explain above, we are unaware of existing literature that intentionally explores and explicitly deals with these issues.
> \
> By identifying the existence of both downward and upward bias, our work delivers a new message: when developing anomaly detection models, we need to treat labeled anomalies carefully at model training time. Ideally one should design training to utilize upward scoring bias while avoiding harmful downward scoring bias. We believe this leads to new opportunities for improving anomaly detection design under a more diverse set of scenarios.

---

> > ### Author Response · Authors · 2020-11-16
> > **Clarifications to questions in problem statement, title, etc.**
> >
> > Below please find our further response (**A**) to the questions regarding the other questions (**Q3-5**).
> >
> > **Q3: [Justification of the problem statement]**
> > (“problem is ill-posed”)
> > \
> > **A**: We realize that the current text in Section 3.1: “We formulate the anomaly detection problem in a (semi-) supervised binary classification setting” could be unclear. Here our use of “binary classification’’ was meant to refer readers to the first subproblem of Eq. (3.2). Our target anomaly detection problem, as formally stated in Eq. (3.1), is in fact different from the classical supervised classification setting. Here the key difference lies in the final performance measure (i.e., TPR under a given constraint on FPR, as opposed to any classification loss). We will clarify this in the revision by removing “binary classification” from the context.
> > \
> > Another related change we plan to make is to explicitly state that our definition of the anomaly detection problem (Eq. (3.1)) aligns with a number of recent works. For example, Li et al. (2019) show that this problem setting is desirable in real-world anomaly detection problems, where engineers hope to detect anomalies with a low false alarm rate (by setting the detection threshold based on a target FPR); Liu et al. (2018) formulate the anomaly detection in a similar way, where the where the goal is to minimize FPR for a fixed recall (TPR)). We will add additional references as well.
> >
> > Does this clarify your concern in the formal statement of the anomaly detection problem?
> >
> > *Reference:*
> > \
> > Li et al. 2019. “Scaling Deep Learning Models for Spectrum Anomaly Detection”. Mobihoc, 2019.
> > \
> > Liu et al. 2018. “Open Category Detection with PAC Guarantees”. ICML, 2018.
> >
> >
> > **Q4: [Clarification on the definition of relative scoring bias]**
> > (“unsupervised anomaly detector as an approximation of the optimal anomaly detector”)
> > \
> > **A**: We did not try to claim “unsupervised anomaly detector” as an approximation of the optimal anomaly detector.  Our definition of “relative scoring bias’’ is to measure the performance difference between *any* two scoring functions. As such, our theoretical results in Section 4 are not restricted to using any specific anomaly detector as the reference scoring function.  We plan to refine our text to clarify this point.
> > \
> > In our empirical result (Section 5), we did compare the results of unsupervised and semi-supervised anomaly detection, and illustrated the existence of upward and downward bias across different classes of test anomalies.  Here we also do not claim that unsupervised anomaly detection is an optimal detector.  Our message here is that semi-supervised anomaly detection (when trained on skewed anomalies) does not always outperform its unsupervised version in all anomaly classes; instead, some classes perform better and some worse. These results confirm the existence of scoring bias in certain training settings, and call for more research efforts in this direction.
> >
> >
> > **Q5: [Title]**
> > (“It should explicitly limit the scope to deep anomaly detection”)
> > \
> > **A**:  Thanks for pointing this out. Yes we will update the title to reflect the focus on deep models:
> > “Understanding Bias in Deep Anomaly Detection: A Semi-Supervised View with PAC Guarantees”

---

> > > ### Author Response · Authors · 2020-11-16
> > > **Clarification of experimental setup and preview of additional experiments**
> > >
> > > **Q6: [Experimental results]**
> > > (“performance of each detector on individual datasets varies significantly” and “standard deviation of these results?”)
> > >
> > > **A**: Following this comment, we will include full (mean, STD) results in our revised version, and are running additional experiments on a broader set of experiment configurations.  The results reported in the current submission are reasonably robust -- as a preview, the following tables listed the (mean, STD) version of Tables 2 and 3.  Here we observe the same conclusion (the presence of downward/upward bias) as the original submission.
> > >
> > > (Updated table 2 & 3 with std reported, 100 runs of experiments)
> > >
> > > |          |  Deep SVDD &nbsp; |  Deep SAD &nbsp;&nbsp;    |     HSC &nbsp; &nbsp;&nbsp;&nbsp;&nbsp;&nbsp; &nbsp;&nbsp; &nbsp;&nbsp;  |     AE&nbsp; &nbsp;&nbsp;&nbsp;&nbsp;&nbsp; &nbsp;&nbsp; &nbsp;&nbsp;&nbsp;&nbsp;     |     SAE &nbsp; &nbsp;&nbsp;&nbsp;&nbsp;&nbsp; &nbsp;&nbsp; &nbsp;&nbsp;   |     ABC &nbsp;      |
> > > |----------|--------------|-------------|-------------|-------------|-------------|--------------|
> > > | shirt    | 0.09 ± 0.01  | 0.71 ± 0.01 | 0.70 ± 0.01 | 0.12 ± 0.01 | 0.72 ± 0.01 | 0.72 ± 0.01  |
> > > | pullover | 0.13 ± 0.02  | 0.90 ± 0.01 | 0.89 ± 0.01 | 0.19 ± 0.02 | 0.84 ± 0.01 | 0.85 ± 0.01  |
> > > | coat     | 0.14 ± 0.03  | 0.92 ± 0.02 | 0.92 ± 0.01 | 0.15 ± 0.02 | 0.92 ± 0.02 | 0.92 ± 0.01  |
> > > | dress    | 0.17 ± 0.03  | 0.24 ± 0.03 | 0.24 ± 0.03 | 0.11 ± 0.01 | 0.20 ± 0.03 | 0.21 ± 0.03  |
> > > | bag      | 0.49 ± 0.07  | 0.38 ± 0.08 | 0.36 ± 0.07 | 0.70 ± 0.03 | 0.52 ± 0.09 | 0.53 ± 0.07  |
> > > | trouser  | 0.32 ± 0.10  | 0.07 ± 0.04 | 0.06 ± 0.03 | 0.59 ± 0.04 | 0.07 ± 0.04 | 0.16 ± 0.07  |
> > > | boot     | 0.92 ± 0.03  | 0.29 ± 0.15 | 0.27 ± 0.16 | 0.98 ± 0.02 | 0.90 ± 0.09 | 0.90 ± 0.08  |
> > > | sandal   | 0.30 ± 0.04  | 0.26 ± 0.08 | 0.26 ± 0.12 | 0.82 ± 0.02 | 0.46 ± 0.10 | 0.56 ± 0.09  |
> > > | sneaker  | 0.55 ± 0.09  | 0.12 ± 0.10 | 0.14 ± 0.12 | 0.74 ± 0.09 | 0.47 ± 0.19 | 0.46 ± 0.18  |
> > >
> > >
> > >
> > >
> > > |                |  Deep SVDD &nbsp; &nbsp;  |  Deep SAD &nbsp; &nbsp;  |     HSC &nbsp; &nbsp;&nbsp;&nbsp;&nbsp;&nbsp; &nbsp;&nbsp; &nbsp;&nbsp;    |     AE &nbsp; &nbsp;&nbsp;&nbsp;&nbsp;&nbsp; &nbsp;&nbsp; &nbsp;&nbsp; &nbsp;&nbsp;      |     SAE &nbsp; &nbsp;&nbsp;&nbsp;&nbsp;&nbsp; &nbsp;&nbsp; &nbsp;&nbsp;     |     ABC      |
> > > |----------------|--------------|-------------|-------------|-------------|-------------|--------------|
> > > | damp grey soil | 0.12 ± 0.05  | 0.81 ± 0.02 | 0.80 ± 0.02 | 0.00 ± 0.00 | 0.08 ± 0.02 | 0.01 ± 0.01  |
> > > | red soil       | 0.67 ± 0.16  | 0.92 ± 0.05 | 0.91 ± 0.05 | 0.00 ± 0.00 | 0.00 ± 0.00 | 0.00 ± 0.00  |
> > > | grey soil      | 0.45 ± 0.17  | 0.92 ± 0.02 | 0.92 ± 0.02 | 0.01 ± 0.00 | 0.04 ± 0.02 | 0.02 ± 0.02  |
> > > | vegetable soil | 0.40 ± 0.10  | 0.19 ± 0.04 | 0.18 ± 0.04 | 0.35 ± 0.01 | 0.35 ± 0.01 | 0.35 ± 0.00  |
> > > | cotton crop    | 0.96 ± 0.06  | 0.80 ± 0.06 | 0.70 ± 0.08 | 0.89 ± 0.02 | 0.90 ± 0.01 | 0.90 ± 0.01  |

---

### Official Review · AnonReviewer3 · 2020-10-30
**Weak accept**

**Rating:** 7
**Confidence:** 4

**Review:**

#### Problem Statement

This paper considers the effect of bias in anomaly detection. The anomaly detection setup is this: A model is trained on a $1- \alpha, \alpha$ mixture of normal and anomalous examples . The model returns an anomaly score $s(x)$ for each point $x$. We then chose a threshold $\tau$ which ensures a bound on the false  positive rate, while maximizing the number of true anomalies that are labelled $1$.

There are two different sources of bias that the paper considers:

1. Bias coming from not having sufficient samples. Here the paper proves theoretical results quantifying the convergence in terms of the quantile functions and experiments confirming these bounds.

2. Bias from the data, where the set of anomalies available at training is not representative of the test distribution of anomalies. This is more of a domain adaptation problem. Here they present experimental results showing that the behavior of algorithms can in some settings be worse than if there were no anomalies at all.


 ### Pros

1. The paper is easy to read and scholarly. The definitions are precise, the literature survey is thorough. The process of choosing a scoring threshold that they describe is folklore, but it is nice that they formalize it and prove guarantees about it.

2. The experiments related to bias in scoring resulting from bias in the training data are interesting. They have a class of normal points("Tops" in FashionMNIST), and experiment with different choices of anomalies, those are are visually  similar (eg. shirts) and those that are dissimilar (eg. shoes). They find that having only "similar" anomalies while training can detract from the ability  to find "dissimilar" anomalies at test, while improving on the ability to find "similar" anomalies. But having "dissimilar" anomalies even at training does not seem to affect other classes adversely.

### Cons


1. The sample complexity bounds are what you would expect given known bounds on the convergence rate of empirical quantiles to the true quantiles.

2. In the experiments, I found the setup to be a little artificial. In particular, in the "similar" setting, when the model is only given "tops" as normal and "shirts" as abnormal, it is not too surprising that the model is good at distinguishing "top"-like things,  but not so good as distinguishing boots from tops. This is an extreme case of data bias where you only have several distinct kids of anomalies but only one kind is available for training.  A more convincing experiment might be to use different mixture weights for the anomaly groups (like boots, shirts, dresses) at train and test time. It would also be nice to use more "real-world" datasets, say from the [ODDS dataset.](http://odds.cs.stonybrook.edu/)

3. The terminology of unsupervised and semi-supervised that this paper uses is not standard (although this lakc of standardization might be the fault of the area in general, rather than this paper in particular). The [survey by Chandola and Banerjee](http://cucis.ece.northwestern.edu/projects/DMS/publications/AnomalyDetection.pdf) uses semi-supervised for what this paper calls unsupervised (where we only have normal examples at training time). Unsupervised there refers to the setting where one has no labels at all. This class would not a have a name under this paper's current taxonomy. Given that the survey has over 8000 citations by now, I strongly suggest the authors adopt that terminology.


### Summary

As I see it, the paper has two distinct contributions: studying the biasing effect of limited samples and skewed data.  While I liked both aspects of the paper, I can't say either by itself is a particularly strong or surprising result. But they fit well together, and bring rigor to an area which could use more of it. I lean towards accepting the paper.

---

> ### Author Response · Authors · 2020-11-16
> **Authors' response to AnonReviewer3 (Clarifications to questions in contribution, experiments, and terminology)**
>
> Thank you for the detailed review and comments. Below please find our response (**A**) to the main questions (**Q1-3**).
>
> **Q1: [Theoretical contribution]**
> (“sample complexity bounds are what you would expect given known bounds on the convergence rate of empirical quantiles to the true quantiles.”)
> \
> **A**:  We agree that at a high-level, our results in Theorem 3 are intuitive based on known bounds.  One wrinkle that is new is that the proof of Theorem 3 relies on a novel adaption of the key theoretical tool from Massart (1990) (see Sec 2), which allows us to cast finite sample complexity bounds for estimating the TPR (objective) under certain FPR (constraints).
>
> **Q2: [Experimental setup]**
> (“setup to be a little artificial”)
> \
> **A**: Thank you for the suggestion!  Following this suggestion, we have conducted initial experiments on the weighted mixture class settings in the Fashion-MNIST dataset, and obtained similar conclusions on these experiments.  Specifically, we ran three configurations of group training (normal: top; abnormal: shirt & sneaker) by varying the weights of the two abnormal classes in training (0.5/0.5, 0.9/0.1, 0.1/0.9). Under these settings, we also identified downward bias for some abnormal test classes, e.g. trousers, and upward bias for other classes.  We plan to add these results and related discussions to the rebuttal revision.
> \
> We are also running additional experiments on other real-world datasets, and plan to report new results in our rebuttal revision soon.  Note that StatLog (used by our submission) is a dataset from the ODDS dataset.
>
> **Q3: [Terminology]**
> (“ terminology of unsupervised and semi-supervised that this paper uses is not standard ”)
> \
> **A**:  We appreciate the suggestion and the pointer to the survey paper. We agree that the term “semi-supervised anomaly detection’’ has been defined differently by existing literature. One option is to change the terms used in our paper as follows:
> * unsupervised anomaly detection  →   semi-supervised anomaly detection (trained on normal)
> * semi-supervised anomaly detection →  semi-supervised anomaly detection (trained on normal + limited abnormal)
>
>
> In this way, we maintain the definition that semi-supervised anomaly detection is only trained on some classes (but not all the classes like supervised learning), and highlight the fact that the choice of training data could lead to model bias.  We also plan to add a table to clearly define these terms and relate them to various existing works, including the survey paper.
> \
> Do you think these changes would help clarify the terminology issue?

---

> > ### Comment · AnonReviewer3 · 2020-11-24
> > **Update**
> >
> > Terminology:
> >
> > Thank you for taking the terminology suggestions seriously. I do worry though that even your new suggestion does risk creating some confusion. Given the Chandola-Banerjee survey is about 10 years old and has close to 10,000 citations, I think the best option is to stick to their terminology, where having both labels is called supervised anomaly detection. In the Chandola-Banerjee nomenclature, your results say that semi-supervised anomaly detection might be better than supervised anomaly detection, if the anomalous labels are biased.
> >
> > I quote from the discussion in section 2.3.1 of the survey
> >
> > *There are two major issues that arise in supervised anomaly detection. First, the anomalous instances are
> > far fewer compared to the normal instances in the training data. Issues that arise
> > due to imbalanced class distributions have been addressed in the data mining and
> > machine learning literature [Joshi et al. 2001; 2002; Chawla et al. 2004; Phua et al.
> > 2004; Weiss and Hirsh 1998; Vilalta and Ma 2002]. Second, obtaining accurate
> > and representative labels, especially for the anomaly class is usually challenging.
> > A number of techniques have been proposed that inject arti¯cial anomalies in a
> > normal data set to obtain a labeled training data set [Theiler and Cai 2003; Abe
> > et al. 2006; Steinwart et al. 2005].*
> >
> > It seems that this discussion sets up your work nicely: you address the problems that arise when the set of anomalous labels are not representative of the true anomalies. But it also clearly includes your setup under the umbrella of supervised anomaly detection, moreover the issues you address are foreseen.
> >
> > Separately, I am happy to see that the results hold even in the mixture setting. In my view this makes the paper stronger, it also brings it closer to work on domain adaptation. But I don't believe that this is "covered by previous work", at least not the prior work that I am aware of. Obtaining representative anomaly labels is challenging, and your work highlights the unexpected downsides to not having truly representative labels. I am increasing my score to 7.

---

### Author Response · Authors · 2020-11-25
**Summary of changes in rebuttal revision**

We thank the reviewers for their detailed comments and valuable suggestions.  We studied the reviews and discussions carefully and modified our paper accordingly.  Our revision followed the same list of actions proposed in our rebuttal responses and further feedback from the reviewers.

We uploaded a [revision version where the changes are highlighted in red color](https://openreview.net/references/pdf?id=EZMz1JP0WL); and also an [unmarked version](https://openreview.net/pdf?id=sAzh_FTFDxz).

Next we summarize the (key) changes included in our revision, for each paper section.

**[Title]**

Following the reviewers’ comments, we have updated the title to “Understanding the Effect of Bias in Deep Anomaly Detection”, to emphasize our focus on deep anomaly detectors.

**[1. Introduction]**

We have clarified and highlighted our contribution in the revised introduction section.

We also added discussions on the terminology used for the anomaly detection regimes considered in this paper, to better position our work in the literature. Specifically, following the reviewers’ suggestion, we switched our terminology on anomaly detection models to the ones used by the Chandola-Banerjee survey paper (w. 10K citations). We added explicit descriptions on these models, and highlighted the fact that existing works have used different terminology for them.

**[2. Related work]**

We further clarified the terminology according to the Chandola-Banerjee survey. We  now use the terms “semi-supervised anomaly detection” and “supervised anomaly detection” with more explicit descriptions according to what data is used in the training process.

**[3. Problem formulation]**

We provided concrete contexts to justify the practical importance of our formulation of the anomaly detection problem. When presenting our problem statement, we now make it clear that it differs from the classical supervised classification setting due to its different performance measure. We also fixed a few editorial issues as pointed out by AnonReviewer1.

**[4. Finite sample analysis]**

We have added a concrete reference to Parzen (1980) when introducing Proposition 1. Furthermore, we have included **additional empirical results to demonstrate our finite sample convergence rate** (Theorem 3) on several **real-world** anomaly detection datasets, including a new dataset on cellular spectrum usage, measured under  normal usage and abnormal usage corresponding to  four types of attacks. We summarized our observations in the main paper, and added the full results (including the anomaly scores distribution and the convergence plots) to Appendix D. These new results align with existing results/observations.

**[5. Impact of scoring bias]**

We have included **additional experimental results on the impact of scoring bias** introduced by different anomaly training sets. To address the earlier concern on our experimental setup, we now include results under more *“realistic scenarios”* on three  real-world anomaly detection datasets: i.e. Fashion-MNIST, Statlog (Landsat Satellite) from the ODDS site, and Cellular Spectrum Misuse from a recent paper).

Following AnonReviewer3’s suggestion, we also evaluated the deep anomaly detectors based on biased anomaly training sets sampled from mixture distributions over multiple categories. We considered this to be a new “Scenario 3” , and reported a consistent pattern of the bias effects in Section 5. The full results are provided in Appendix E.

Following AnonReviewer2’s suggestion, we have now reported both mean and standard deviation for all our results (see all the tables in Appendix E).

Following AnonReviewer4’s suggestion, we added text in the end of this section to interpret why the model behaviors of scenario 1 and 2 are different.

**[6. Conclusion]**

We have revised our discussion on the key messages of this work as explained in our rebuttal, to highlight the significance of this work.

---

### Decision · Program_Chairs · 2021-01-07
**Final Decision**

**Decision:**

Reject

**Comment:**

This paper studies the effect of anomaly detection using supervised learning with non-representative abnormal examples on the TPR of the anomaly detection model. Experiments demonstrate that when the abnormal examples presented in the training set are not representative of the abnormal examples in the target distribution, this can lead to bias in estimating the TPR.

Pros:
- This paper considers an important issue of tuning anomaly detection models in the absence of representative anomalies.
- The paper is well written and easy to read.
Cons:
- The analysis and experiments provided in the paper are not surprising, and repeat, although perhaps in more detail, known effects of learning with a non-representative training sample.